

# Competition between Chaotic Advection and Diffusion:
# Stirring and Mixing in a 3D Eddy Model

Genevieve Jay Brett[1], Larry Pratt[2], Irina Rypina[2], and Peng Wang[3]

[1]IPRC, University of Hawaii Manoa, Honolulu, HI, USA
[2]Woods Hole Oceanographic Institution, Woods Hole, MA, USA
[3]University of California, Los Angeles, CA, USA

*Correspondence to:* Genevieve Jay Brett (brett33@hawaii.edu)

**Abstract.**

   The importance of chaotic advection relative to turbulent diffusion is investigated in an ideal­ized model of a 3D swirling and overturning ocean eddy. Various measures of stirring and mixing are examined in order to determine when and where chaotic advection is relevant. Turbulence is

alternatively represented by: 1) an explicit, observation–based, scale–dependent eddy diffusivity, 2) stochastic noise, added to a deterministic velocity field, or 3) explicit and implicit diffusion in a spec­tral numerical model of Navier–Stokes equations. Lagrangian chaos in our model occurs only within distinct regions of the eddy, including a large chaotic 'sea' that fills much of the volume near the perimeter and central axis of the eddy, and much smaller 'resonant' bands. The size and distribution

of these regions depends on factors such as the degree of axial asymmetry of the eddy and the Ek­man number. The relative importance of chaotic advection and turbulent diffusion within the chaotic regions is quantified using three measures: the ratio of the tracer filament arrest scale to the width of the chaotic region, the rate of dispersal of closely spaced fluid parcels, and the Nakamura effective diffusivity. The role of chaotic advection in the stirring of a passive tracer is generally found to be

most important within the larger chaotic 'seas', at intermediate times, with small diffusivities, and for eddies with strong asymmetry. In contrast, in thin chaotic regions, turbulent diffusion at oceano­graphically relevant rates is at least as important as chaotic advection. Future work should address anisotropic and spatially–varying representations of turbulence for more realistic models.



## 1   Introduction

Chaotic advection (Aref (1984); Shepherd et al. (2000)) is a process by which rapid stirring, as manifested by the stretching and folding of material, is produced within a smooth and well organized Eulerian velocity field. The enhancement of stirring can be attributed to chaotic fluid parcel trajectories and their rapid separation from nearby trajectories. There are many examples, ranging from simple models of purely laminar flow (e.g. Rom-Kedar et al. (1990); Samelson (1992); Pierre-

humbert (1994); Malhotra et al. (1998); Poje and Haller (1999); Coulliette and Wiggins (2001) and other work reviewed in the texts of Ottino (1990); Samelson and Wiggins (2006)), to modeled or observed, oceanographically or atmospherically relevant flows (e.g. Rogerson et al. (1999); Miller et al. (2002); Deese et al. (2002); Olascoaga and Haller (2012); Sayol et al. (2013); Rypina et al. (2007, 2009, 2011a, 2012)). In most cases the flow fields are two–dimensional and time–dependent,

and when observed, often occur at the sea surface or within the stratosphere (Polvani et al. (1995); Ngan and Shepherd (1997)). Three dimensional examples also exist (e.g. Fountain et al. (2000); Rypina et al. (2015); Solomon and Mezić (2003); Yuan et al. (2004); Branicki and Kirwan Jr (2010), and Pratt et al. (2014), hereafter P2014) and often involve numerically modeled velocity fields, due to the limitations of observational methods.

35       A feature that is intriguing and quite common in these studies is that Lagrangian chaos is confined to certain sub–regions of the flow field, separated from each other by bands of material curves or surfaces that contain no chaotic Lagrangian motion. The chaotic regions are rapidly stirred as a result of the signature rapid separation of nearby trajectories, but the non–chaotic bands act as barriers that confine the stirring. In textbook examples, including area–preserving maps of time–periodic 2D or

steady 3D velocity fields, the chaotic and non–chaotic regions form a fractal geometry, with bounded chaotic regions imbedded in larger chaotic seas, themselves bounded and imbedded in even larger chaotic regions (Chirikov (1971, 1979); Casati and Ford (1979); Gromeka (1881); Dombre et al. (1986)). In finite–time systems or systems with arbitrary time dependence, the distinction between chaotic and regular trajectories is difficult to define. A great deal of recent work in the field has

resulted in the development of methods for identifying material barriers based on the notion of Lagrangian coherence. These methods include, for instance, finding sets of trajectories that experience fastest separation rates from their close neighbors, identifying contours that undergo minimal stretching, locating sets of trajectories that remain compact in some sense and/or share a common property, or identifying trajectories that encounter the largest number of other trajectories (see Haller (2002);

Shadden et al. (2005); Froyland et al. (2007, 2012); Rypina and Pratt (2017); Rypina et al. (2018); Hadjighasem et al. (2017); Rypina et al. (2011b); Haller and Beron-Vera (2012, 2013) as well as the review by Haller (2015) and references contained therein). Applications of these methods often result in the identification of material contours and surfaces that act as barriers over finite time, thus allowing for partitioning between strongly and weakly stirred regions of the flow field.



Completely impenetrable material barriers only exist because of the deterministic nature of the trajectories. Even a low level of background turbulence at small scales, if represented as a diffusive process, would cause the barriers to become permeable or fuzzy over sufficiently long periods of time, and perhaps nonexistent in any practical sense if the time scale of interest is long enough. The relevance of chaotic advection for the stirring of material within geophysical flows would appear to

rest on several criteria. The first is that the flow field contain persistent, long–lived (on the time–scale of interest) features such as gyres, eddies and jets, that by themselves generate regions of elevated stirring as well as separating barriers. Secondly, the stirring within these regions should be at least as important as that due to smaller scale, intermittent features (i.e. small scale turbulence). Third, the barriers that exist in the absence of small–scale turbulence should retain meaning as suppressors

of exchange between the rapidly–stirred regions in the presence of the small–scale turbulence. For the flow considered in this paper the first aspect has been investigated and shown to be true (P2014; Rypina et al. (2015)); this work concentrates on investigating the second and third aspects.

The terms "important" and "relevant" are somewhat subjective, and a particular aspect, such as the existence of barriers, that is of interest to one person may not be so to another person. We ex-

amine several measures of stirring and mixing in a particular case of a three–dimensional flow field: an idealized representation of an isolated eddy with horizontal swirl and vertical overturning. This idealized eddy is most likely to be similar to a submesoscale eddy within a surface mixed layer of the ocean, although the velocities of such eddies have not been well observed. The effects of stirring and mixing at these smaller scales, where vertical velocities become important, is increasingly un-

der study (e.g. Mahadevan (2016)). Generally, increased resolution improves ocean model behavior (Griffies et al. (2015)), so at lower resolutions, an ongoing challenge is parameterizing sub–grid–scale processes (e.g. Hallberg (2013)).

Our three–dimensional flow contains Ekman layers at the top and bottom of a cylindrical domain and their thickness relative to the full depth is measured by an Ekman number. The Lagrangian struc-

ture of the steady as well as time–periodic, deterministic versions of this flow has previously been explored (P2014; Fountain et al. (2000); Rypina et al. (2015)). This deterministic flow field can be approximated by an analytically described velocity field (Sect. 2), favorable for the efficient calculation of large numbers of trajectories. In this paper, we will add a stochastic disturbance representing small–scale turbulence to the deterministic flow. In addition, some of our calculations are done using

velocity fields from a direct numerical integration of the Navier–Stokes equations (used in Sect. 5).

In order to examine the relevance and importance of stirring and mixing due to large–scale Lagrangian chaos compared to that due to small–scale turbulence, we use several distinct measures applied to our isolated eddy model. The first measure is a Lagrangian version of the Batchelor scale (Sect. 3), a measure of the smallest tracer filament width that can be produced by chaotic advection

before small scale–diffusion arrests the progression to smaller scales. The second measure (Sect. 4) involves the dispersion of ensembles of initially closely spaced trajectories. The final measure (Sect.



5) is a bulk or "effective" diffusivity (Nakamura (1996)) that indicates the rate of irreversible mixing between volumes with different tracer concentrations. The analyses in Sections 3–4 are based on a "kinematic" analytical model with and without stochastic perturbation; the analysis in Sect. 5 is
based on a "dynamical" numerical solution of the Navier–Stokes equations.

## 2   Models

We will consider the steady flow of a homogeneous and incompressible fluid in a rotating cylinder of depth H, driven at the top by the stress due to a differentially rotating lid. The resulting circulation has Ekman layers at the top and bottom, and thus a central parameter is the Ekman number

$$E = \left( \nu/\Omega H^2 \right) = \left( \delta_E/H \right)^2 ,  \tag{1}$$

where $\nu$ is the kinematic viscosity, $\Omega$ is the angular rate of rotation of the cylinder, and $\delta_E$ is the thickness of the Ekman layers. Much oceanographic literature has been devoted to the case in which the differential lid rotation $\delta\Omega$ is small $(\delta\Omega/\Omega) \ll 1)$, and the Ekman layers are relatively thin, $E \ll 1$. In this case a linear, asymptotic solution is available (Greenspan (1968) and Appendix A

of P2014). According to this solution (with $\delta\Omega > 0$) , fluid is drawn up into the top Ekman layer from an inviscid and vertically rigid interior region that rotates at half the angular velocity of the lid. The fluid is carried radially outward and then downward within thin, viscous side–wall layers. When it reaches the bottom, the fluid flows radially inward in a bottom Ekman layer and expelled upward into the interior region. Fluid trajectories thus spiral upwards in the interior, outwards in the

top Ekman layer, downwards near the side walls, and inward in the bottom Ekman layer.

Although the set–up described above and its linear asymptotic treatment have provided a foundation for a wide variety of models with geophysical and industrial applications (e.g. Lopez and Marques (2010)), it is not the most convenient for Lagrangian studies. One difficulty is that all fluid trajectories pass through small corner regions at the top and bottom of the cylinder. These regions

are not resolved by the asymptotic solution and can be difficult to resolve numerically, particularly when the velocity field is to be used to accurately calculate trajectories that are cycling through the cylinder numerous times. For this reason it is advantageous to modify the forcing at the upper surface to conform to a stress that still acts in the azimuthal direction and is zero at the cylinder axis, but approaches zero at the cylinder boundary as well. P2014 used one such forcing distribution to

create a flow in which the downwelling occurs over a broad outer region of the inviscid interior, no longer confined to the thin, viscous sidewall layers. We will use the same velocities (obtained from a numerical model) for the tracer release experiments discussed later in this work.

Since numerical solutions are required to get a complete, dynamically consistent velocity field for the rotating cylinder, Lagrangian calculations requiring long integration times can become cumber-

some, making it difficult to explore the variations in the governing parameters. As a compromise, past investigators have developed phenomenological models in which an incompressible Eulerian



velocity field containing the qualitative features of the dynamically consistent fields is specified analytically and fluid trajectories are computed from it. Many of the calculations described below are based on such a model, hereafter referred to as the "kinematic" model. This new model is an im-

provement on the phenomenological model used by P2014 and Rypina et al. (2015) in terms of its more realistic portrayal of Ekman layers and inclusion of the Ekman number as a parameter.

The kinematic model specifies an analytically prescribed background velocity field that is steady, incompressible, and has no azimuthal structure. Under these conditions, all trajectories are regular, or non–chaotic. When perturbed through the addition of an analytically prescribed symmetry–

breaking disturbance, one with azimuthal structure, Lagrangian chaos arises in portions of the three–dimensional flow field. The velocity field is specified in nondimensional cylindrical coordinates $(r, \theta, z)$, with $(1 \geq z \geq 0)$ and $(r \leq a)$, where $a$ is the width–to–height ratio of the domain. The background flow has $\partial/\partial\theta = 0$ and can be expressed as the sum of an azimuthal velocity $V(r, z)$ and an overturning circulation with radial and vertical velocity components $U(r, z)$ and $W(r, z)$.

The latter are specified by the streamfunction

$$\Psi = -E^{1/2} R(r) F(z), \tag{2}$$

where $F(z)$ is the vertical portion of the streamfunction, and $R(r)$ is the radial portion of the streamfunction. The vertical portion of the streamfunction is

$$F(z) = A[\sin(\zeta)\sinh(\zeta) - \cos(\zeta)\cosh(\zeta)] + B[\sin(\zeta)\sinh(\zeta) + \cos(\zeta)\cosh(\zeta)] - D, \tag{3}$$

where $\zeta$ is a transformed vertical coordinate,

$$\zeta = \frac{z - 1/2}{E^{1/2}}, \tag{4}$$

and the constants are defined by

$$A = \frac{-1}{2}\frac{cS}{s^2C^2 + c^2S^2}, \quad B = \frac{1}{2}\frac{sC}{s^2C^2 + c^2S^2}, \quad D = A(sS - cC) + B(sS + cC)$$

$$s = \sin\left(\frac{1}{2E^{1/2}}\right), \quad c = \cos\left(\frac{1}{2E^{1/2}}\right), \quad S = \sinh\left(\frac{1}{2E^{1/2}}\right), \quad C = \cosh\left(\frac{1}{2E^{1/2}}\right). \tag{5}$$

In the limit of infinite cylinder radius, $a \to \infty$, the radial portion of the streamfunction, $R(r) = r^2/s$, yields a dynamically consistent solution for flow between two differentially rotating, horizontal plates. Fluid flows radially inward within the bottom Ekman layer and is expelled upward and eventually into the top Ekman layer, where it moves radially outward. When $a$ is finite the velocity needs to vanish at the cylinder walls, and this can be accomplished by choosing $R$ as

$$R(r) = r(a - r)^2/2, \tag{6}$$

giving velocities

$$U = \frac{-\partial\Psi}{\partial z} = r(a - r)^2[A\sin(\zeta)\cosh(\zeta) + B\cos(\zeta)\sinh(\zeta)], \tag{7}$$

$$W = \frac{1}{r}\frac{\partial r\Psi}{\partial r} = -(a - r)(a - 2r)E^{1/2}F(z) \tag{8}$$



where $U$ is radial and $W$ is vertical.

The axisymmetric azimuthal velocity $V$, satisfying the incompressibility condition in 3D, is defined as

$$V(r,z) = r(a-r)^2 [\frac{1}{2} + B\sin(\zeta)\cosh(\zeta) - A\cos(\zeta)\sinh(\zeta)]. \tag{9}$$

This velocity leads to typical nondimensional trajectory rotation times of 18–200 for all Ekman numbers examined; the central orbit at $(r,z) = (0.5, 0.5)$ has a period of $16\pi \approx 50$. Model horizontal

velocities are typically between $0.01$ and $0.1$ in magnitude, which are reasonable ocean velocities in meters per second. This choice of the velocity scale being $1\mathrm{ms}^{-1}$ gives rotation times of several hours assuming the eddy radius is equal to its height ($a = 1$). Using the same scaling for vertical velocities, whose nondimensional values are $E^{1/2}$ smaller, gives overturning times of 7 hours to 2 months; although eddies with this structure have not been carefully observed, vertical velocities near

submesoscale fronts reach $30\mathrm{m}\,\mathrm{day}^{-1}$, which is in line with these rates. For all parameter values, there is upwelling in the center ($r = 0$) and weaker downwelling near the sides of the cylinder (strongest at $r = 0.75a$). There is horizontal convergence near the bottom and divergence near the top; for $E$ near one, these are true for the full bottom and top halves of the system.

As the Ekman number varies, the overturning streamfunction changes qualitatively (Fig. 1). For

$E > 1/60$ the overturning circulation is rounded and has a single internal fixed point corresponding to the horizontal, circular trajectory described above as the central orbit (Fig. 1a,b). For $E < 1/60$ additional fixed points in the overturning circulation arise at r=0.5 (Fig. 1c). These fixed points in Fig. 1c are again circular periodic trajectories in 3D, and the increasing number arise through pitchfork bifurcations as $E$ decreases (see appendix A for more details). The additional circular trajectories are

associated with smaller overturning cells imbedded in the larger cell (detailed example in appendix A, Fig. 13). The overturning streamfunction also exhibits more vertical rigidity as E decreases, analogous to deeper oceanic columns, in accordance with the Taylor–Proudman Theorem (Greenspan (1968)).

### 2.1    Symmetry–breaking Perturbation

In the kinematic axially symmetric analytically prescribed background flow described above all trajectories move along toroidal surfaces and are thus non–chaotic. In order to use this system to study the interplay of chaotic advection and turbulent diffusion, we must perturb the system to break the axial symmetry, which will introduce chaotic trajectories. The applied perturbation, approximating the flow produced by a lid rotating off–center, is a horizontal flow that decays in strength with depth

and is described by the streamfunction:

$$\widetilde{\Psi} = \epsilon \frac{-\sinh(z/E^{1/2})}{2\sinh(1/E^{1/2})}(a^2 - r^2)(\gamma^2 a^2 - s^2), \; s = \sqrt{(x-x_0)^2 + y^2}. \tag{10}$$

This general form allows for an $r-$ and $z-$ dependent adjustment to the strength of the azimuthal velocity, with amplitude $\epsilon$, and a symmetry breaking component governed by the offset parameter


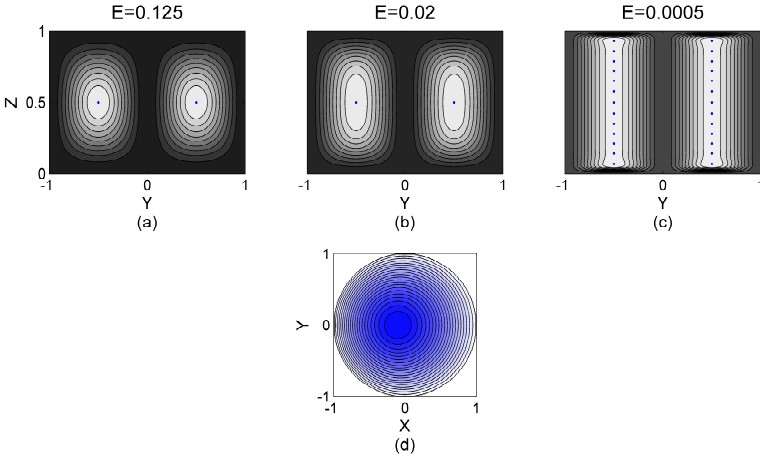

**Figure 1.** Top: Background overturning streamfunction for $a = 1$ Left to right: overturning $E = 0.125$, $E = 0.02$, $E = 0.0005$. Blue dots are $rz$–fixed points. Bottom: horizontal perturbation streamfunction for $\gamma = 2$, $x_0 = -0.5$. Note that the center of rotation in the perturbation streamfunction is not at the origin.

$x_0$. If $x_0 = 0$, the disturbance is axially symmetric; if it is nonzero, the disturbance has an azimuthal

variation of amplitude $\epsilon x_0$. The parameter $\gamma$ can be used to make adjustments in the radial structure of the disturbance. This streamfunction is for velocities in the $x$ and $y$ directions, unlike $r-$ and $z-$ dependent background overturning streamfunction; the velocities from the two are added together. The perturbation velocities in $x$ and $y$ are

$$\tilde{u} \quad = \partial\widetilde{\Psi}/\partial y = 4y\epsilon\frac{\sinh(z/\sqrt{E})}{\sinh(1/\sqrt{E})}\left[(a^2 - r^2) + (\gamma^2 a^2 - s^2)\right], \tag{11}$$

$$\tilde{v} \quad = -\partial\widetilde{\Psi}/\partial y = -4y\epsilon\frac{\sinh(z/\sqrt{E})}{\sinh(1/\sqrt{E})}\left[(x - x_0)(a^2 - r^2) + x(\gamma^2 a^2 - s^2)\right]. \tag{12}$$

The corresponding azimuthal and radial velocity perturbations are

$$\tilde{V} = -2\epsilon\frac{\sinh(z/\sqrt{E})}{\sinh(1/\sqrt{E})}\left[(a^2 - r^2) + (\gamma^2 a^2 - s^2) - \frac{x_0}{r}\cos(\theta)(a^2 - r^2)\right], \tag{13}$$

$$\tilde{U} = 2\epsilon x_0\frac{\sinh(z/\sqrt{E})}{\sinh(1/\sqrt{E})}\sin(\theta)(a^2 - r^2). \tag{14}$$

The perturbation streamfunction's overall strength decays with depth and goes to 0 at the bottom ($z = 0$). For the rest of the work, we will use $a = 1$ and $\gamma = 2$ (Fig. 1d). We note that the total, i.e., background plus perturbation, azimuthal velocity can be zero at some locations in the domain for certain choices of $\epsilon$, but with $\epsilon < 0.05$ these locations are all very close to the boundaries of the cylinder.





### 2.2 Comparison to Dynamic Model

In this section we compare our kinematic model to the Navier–Stokes (NS) simulation of a rotating cylinder flow by P2014. We will use the kinematic model for the analyses in sections 3.1 and 3.2, and the NS simulation for the analysis in Sect. 3.3. We are interested in comparing the qualitative features of the two model flows under steady symmetry–breaking perturbation. It is important to note that the parameters of the two systems are slightly different. The parameters that arise in the NS simulation are the Ekman number, $E$, the aspect ratio, $\alpha$, the displacement $X_0$ of the lid's center (not to be confused with $x_0$ in the kinematic model), and the Rossby number, $Ro = \delta\Omega/\Omega$. The kinematic model parameters are the Ekman number, $E$, the aspect ratio, $a$, the perturbation offset parameter, $x_0$, and the strength of the perturbation, $\epsilon$. For matching the kinematic model to the NS simulation, we set $\alpha = a = 1$ and examine four Ekman numbers used in P2014, $E \in \{0.25, 0.125, 0.02, 0.0005\}$. The displacement and strength of the kinematic perturbation are adjusted to match the behavior for a given Rossby number and displacement of the lid in the dynamic simulation. The chosen values are maintained throughout the rest of the work unless otherwise noted. We do this rather than attempting a mathematical equivalence because the kinematic perturbation has a different form than that describing a physical lid rotating off–center. Our model mimics a flow with a small Rossby number, so we compare our results to those from P2014's $Ro = 0.2$, with lid displacement $X_0 = -0.02$.

Figures 2–3 show some examples of Poincaré maps from the NS simulation (top rows) with maps from the kinematic model (bottom rows). It is important for our purposes to achieve qualitative agreement in terms of the depth of the Ekman layers, the vertical rigidity of the interior regions, and the overall layout of regular, chaotic, and resonant regions. For the choice of the parameters described above, there is a good match of these qualitative features. Each case is marked by the presence of a substantial chaotic region that extends from the radial center around the top and bottom boundaries and to our largest radii near the perimeter of the cylinder. We henceforth refer to this region as the "chaotic sea". Also, in all cases there are many more points near the surface than near the bottom; this is due to the higher azimuthal velocities near the surface, and is seen in both the dynamic and kinematic model. In $E = 0.25$, both Poincare sections show a series of nested closed curves centered around $(r, z) = (0.5, 0.5)$ corresponding to quasiperiodic trajectories on nested tori. Between these are some thin resonant layers with high numbers of small islands. For $E = 0.125$, the main feature is a series of larger islands between a set of nested tori and the chaotic sea. For $E = 0.02$, there is one large island with a number of resonant layers surrounding it, including small islands. For $E = 0.0005$, the vertical structure of both models is more rigid, the kinematic model more so than the NS simulation. Altogether, the kinematic model reproduces the general features of the NS simulations, through there are often differences in details such as the number and widths of islands.


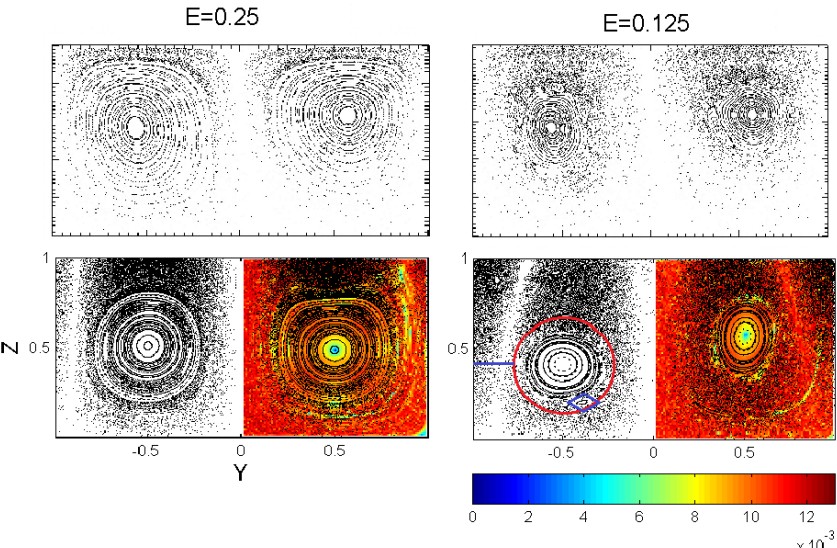

**Figure 2.** Structures in the kinematic model and dynamical simulation for Ekman numbers of 0.25 and 0.125. Top: Poincaré maps from Pratt et al. (2014) (their Fig. 10), using the dynamic simulation. Bottom: in black, Poincaré maps from the current kinematic model with $\epsilon = 0.01$ and $x_0$ either $-0.5$ (left) or $-0.9$ (right); in color, maximum FTLEs calculated for the kinematic model with integration time 400. For $E = 0.125$, red oval approximately separates the resonant and regular layers (inside) from the chaotic sea region (outside), with the blue line segment showing the width of the chaotic sea. The blue diamond shows the width of an island, which is also the width of the resonant layer.

## 3 Lagrangian Batchelor Scale

We examine the relative importance of chaotic advection and eddy diffusion for tracer distribution using three types of methods. We begin with scaling arguments: a Lagrangian Batchelor scale defines the thinnest filaments that can form based on the balance between advection and diffusion. At scales

above the Batchelor scale, advection dominates over diffusion, and vice versa. If the Batchelor scale is wider than the chaotic layer thickness, then we conclude that diffusion is the dominant process in that chaotic layer.

To relate our dimensionless kinematic model variables to ocean eddies, we need to set dimensional length and velocity scaling factors. The main parameter of the background model is the Ekman

number, the square of the ratio of Ekman layer thickness to eddy depth. Due to the unstratified nature of our flow, we focus on two intermediate Ekman numbers: $E = 0.125$ and $E = 0.02$. Assuming an Ekman depth of about 40m, which is within the range of open–ocean observations (see Lenn and Chereskin (2009) and references therein), our shallower eddy is about 110m deep, whereas $E = 0.02$


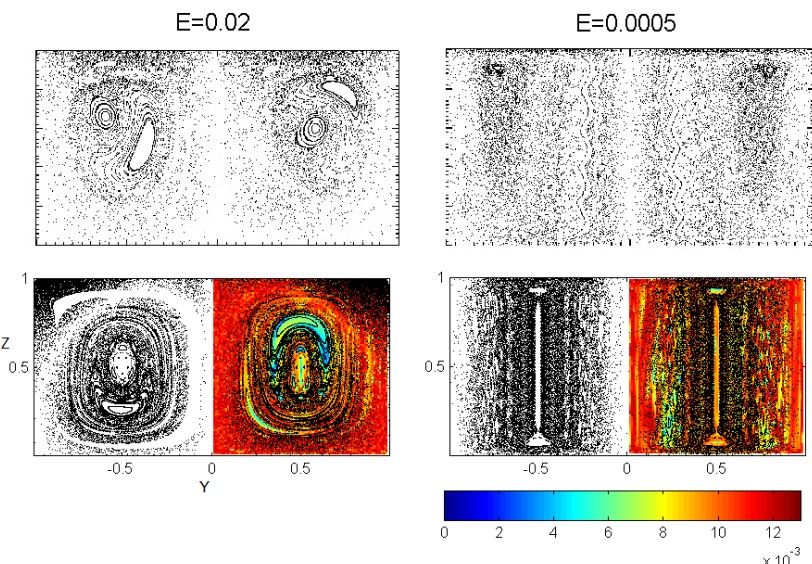

**Figure 3.** Structures in the kinematic model and dynamical simulation for Ekman numbers of 0.02 and 0.0005, same format as Fig. 2.

would correspond to an eddy depth of about 280m. Depending on region and season, it is possible for

either of these to be within the surface mixed layer of the ocean, which can reach 500m in subpolar regions in the winter, but may decrease to a few meters in the summer. Since the aspect ratio of the width–to–depth of our eddy is 1, the corresponding eddy radius is also between roughly 100 and 300m.

   The axisymmetric azimuthal velocity in (10) has a maximum at $r = a/3$, which gives the winding–

around–the–eddy length $2\pi a/3 = 200$–600 m. Dimensionless model velocities are typically between 0.01 and 0.1 in magnitude. To dimensionalize these velocity values, a velocity scaling factor must be chosen. Unfortunately, the background model has only one parameter, which we already used to dimensionalize the eddy depth. So the choice of the velocity scaling factor is arbitrary. However, choosing the velocity scaling factor of 1 ms$^{-1}$, the resulting model azimuthal velocities of 0.01–

0.1 ms$^{-1}$ seem reasonable for the horizontal ocean velocities associated with eddies with length scales of a few hundred meters. This choice gives horizontal rotation, or winding, times around the eddy between 0.5 and 16 hours dimensionally, or 20–210 nondimensionally (for these two Ekman numbers). This relationship also gives a timescale of 100–300s per nondimensional timestep.



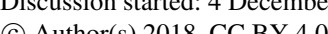



### 3.1 Scaling Derivations

Chaotic advection thins tracer patches through exponential contraction in the contracting direction(s), decreasing the relevant lengthscale towards small scales where diffusion is dominant. Diffusion widens tracer patches by moving tracer down its gradient, spreading it out from its maximum. The length scale at which advection and diffusion balance in their respective thinning and widening of a patch of tracer is the Batchelor scale, $\delta$. Below $\delta$, diffusion dominates tracer behavior, while

above $\delta$ advection dominates. If $\delta$ is larger than the structures in the flow induced by chaos, then diffusion will overcome advection and wipe out these structures. The structures of interest, induced by the deterministic, symmetry–breaking perturbation (see Fig.s 2–3) are the bands of chaos, called resonant layers, surrounding regular island chains (see blue diamond in Fig. 2, lower right), and the chaotic sea region (outside the red oval in Fig. 2, lower right) located near the cylinder perimeter and

central axis.

In principle, the width of a tracer filament should approach the Batchelor scale regardless of initial conditions. If we consider an initial patch of tracer that is far from the Batchelor scale, advection and diffusion will not balance. If the patch is larger than the Batchelor scale, chaotic advection exponentially constricts the patch in the direction of fastest contraction so that it approaches the

Batchelor scale. If the patch of tracer is smaller than the Batchelor scale, diffusion widens the patch to approach the Batchelor scale. When the width of a filament is at the Batchelor scale, the width will be steady in time but the concentration will continue falling.

Traditional formulations of the Batchelor scale use the Eulerian quantity — strain rate — to quantify advection and to find the scale at which advective and diffusive effects balance. Several rigorous

derivations of a Lagrangian Batchelor scale have been presented (eg. Thiffeault (2004); Fereday and Haynes (2004); Son (1999)), and a few papers have used less rigorous scaling arguments to estimate the importance of chaotic advection (Rypina et al. (2010); Ledwell et al. (1993, 1998)). Below we present a simple explanation for the Lagrangian Batchelor scale to gain intuition about this quantity, followed by a rigorous derivation of a Lagrangian Batchelor scale for a for a Gaussian tracer in a 3D

linear strain flow. The latter extends the work of Flierl and Woods (2015) from 2D to 3D.

The first formulation of the Lagrangian Batchelor scale uses dimensional arguments to construct a quantity that has units of $[length]$ from the diffusivity $\kappa$, which quantifies the intensity of diffusion and has units of $[length^2 time^{-1}]$, and the exponential contraction rate $\lambda_3$, which quantifies the thinning of a filament due to chaotic advection and has units of $[time^{-1}]$:

$$\delta = \sqrt{\kappa/|\lambda_3|}. \tag{15}$$

In a flow field with uniform steady strain, one could simply use the Eulerian strain rate as the filament thinning rate. However, in flows with non–constant strain rate, the tracer will feel different strain as it is advected by the flow so a Lagrangian quantity such as the Finite Time Lyapunov Exponent (FTLE) would be more appropriate. The FTLE quantifies the exponential separation rate





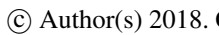

between a trajectory and its close neighbors over time interval $\Delta t$,

$$\Delta x = \Delta x_0 e^{\lambda \Delta t}. \tag{16}$$

Since separation rates between trajectories are generally different in different directions, in 3D flows there are 3 FTLEs that can be ordered $\lambda_1 \geq \lambda_2 \geq \lambda_3$ and can be though of as the stretching/contraction rates of the 3 major axes of an infinitesimal spherical blob of fluid as it deforms

into an ellipsoid under the influence of the flow field (see Fig. 4). For incompressible flows, $\lambda_1 \geq 0$, $\lambda_3 \leq 0$ and $\lambda_1 + \lambda_2 + \lambda_3 = 0$. For the Batchelor scale in eq. (16), the appropriate FTLE is that for the most contracting direction, i.e., $\lambda_3$. FTLEs are most commonly computed as

$$\lambda_i = 1/|T| \ln \sqrt{\sigma_i} \tag{17}$$

where $\sigma_i$ are the eigenvalues of the right Caushy–Green deformation tensor

$$G = [\Delta x_i / \Delta x_0 j]^T [\Delta x_i / \Delta x_0 j]. \tag{18}$$

Here $\Delta x_i$ and $\Delta x_{i0}$ are the final and initial displacements in the i–th direction between initially nearby particles that are advected by the flow over time interval $\Delta t$. $G$ can be calculated directly from dense grids of simulated Lagrangian trajectories. We use this latter method in our calculations to estimate $\lambda_3$.

As an alternative motivation of the Lagrangian Batchelor scale, we show analytically that the width of a Gaussian tracer distribution asymptotically approaches the Batchelor scale in a simple flow field. This derivation is an extension to three dimensions of a two–dimensional calculation by Flierl and Woods (2015). The main steps of the derivation are described below, with more details in the Appendix B. First, we assume that in the Lagrangian frame the velocity field is a steady linear

strain with rates $\lambda_i$ in each direction, such that the sum of the $\lambda$ is zero, giving an incompressible flow. Second, we assume that the tracer concentration $C$ initially has a Gaussian distribution in each direction, and we look for a solution to the tracer evolution equation where it remains Gaussian. In this case we can use the standard deviation of the Gaussian distribution to measure the width of the filament in each direction. The width in the most–contracting direction, which is shrinking with rate

$\lambda_3$, is denoted by $\sigma$. As shown in the Appendix, the differential equation for $\sigma$ has a fixed point at

$$\sigma = \sqrt{\kappa/|\lambda_3|}, \tag{19}$$

meaning that the width of the Gaussian patch in the fastest contracting direction has a fixed point at the Batchelor scale, as expected from the physical arguments about the balance between advection and diffusion. This fixed point is attracting, meaning that for any initial width, the width in the $\lambda_3$

direction will converge to the Lagrangian Batchelor scale. Mathematically there are also fixed points with negative $\lambda_3$ and with negative $\sigma$ for positive $\lambda_3$, but neither corresponds to a real positive tracer distribution. The full solution for $\sigma$ is

$$\gamma = \sqrt{|\lambda_3|/\kappa} \left( (\lambda_3 \gamma_0^{-2}/\kappa - 1) e^{2\lambda_3 t} + 1 \right)^{1/2}. \tag{20}$$




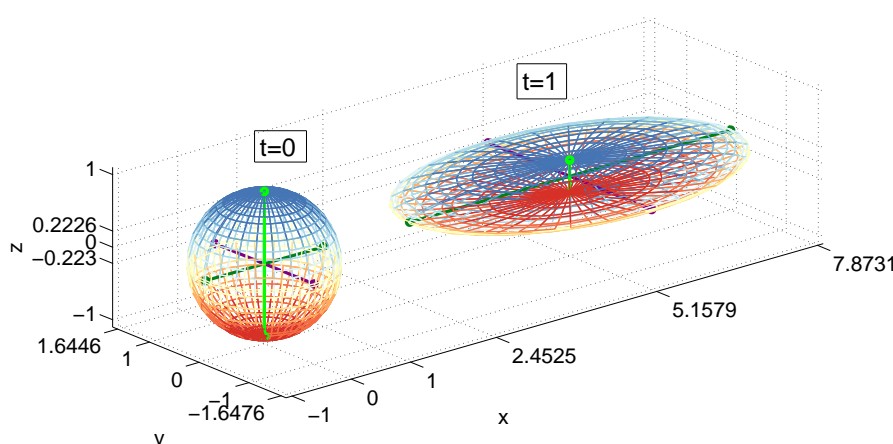

**Figure 4.** An initial sphere in a linear strain field evolving into an ellipsoid during a time of 1. Ellipsoid axes marked by bars, with figure axes ticks showing their endpoint values. Velocity field $u = 1.5 + x$, $v = 0.5y$, $w = -1.5z$. Color shows $z$ values at $t = 0$.

More details and the full solution for $C$ are in Appendix B.

## 3.2 Results of Batchelor Scale Analysis

In order to calculate the Lagrangian Batchelor scale, $\delta$, we use the oceanic diffusivity estimates from Okubo (1971). In the ocean, diffusivity is scale–dependent, increasing with size, as described by Okubo. He used observations of horizontal dye diffusion at various scales ranging between about 20m and 2000km to find the empirical relationship

$$\kappa = 0.0103 l^{1.15}, \tag{21}$$

where $l$ is the horizontal lengthscale of the dye patch in cm and $\kappa$ is in cm$^2$s$^{-1}$. Consistent with the lack of density stratification in our model, we assume an isotropic three–dimensional diffusivity. This assumption is supportable in the upper ocean mixed layer and is consistent with our assumption of shallow eddies.

The variable nature of Okubo's $\kappa$ makes determination of the Batchelor scale a bit more subtle. In the case of spatially variable $\kappa$, the thinning of an initially large tracer patch will occur as before, but as the filaments decrease in width, the corresponding $\kappa$ decreases as well. Following Rypina et al. (2010), we hypothesize that equilibration will occur if during this process the tracer scale $L$



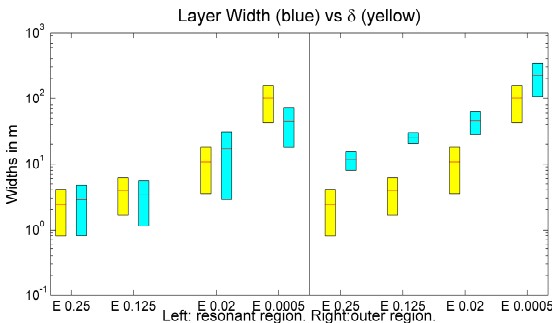

**Figure 5.** Layer widths in blue, Lagrangian Batchelor scale $\delta$ in the same region in yellow. Left half, chaotic resonant region between islands; right half, the chaotic sea region. The diffusivities at the Batchelor scale in m$^2$s$^{-1}$ are between $10^{-4}$ and $6 \cdot 10^{-3}$ for the three larger Ekman numbers and between $1 \cdot 10^{-2}$ and $6 \cdot 10^{-2}$ for $E = 0.0005$.

approaches $(\kappa(L)/|\lambda_3|)^{1/2} = (0.0103L^{1.15}/|\lambda_3|)^{1/2}$. Solving for $L$ yields the Batchelor scale

$$\delta = 0.0046|\lambda_3|^{-1.1765} \tag{22}$$

where $\lambda_3$ in s$^{-1}$ yields $\delta$ in cms$^{-1}$.

The calculated $\delta$ values are shown in Fig. 5 next to the widths of chaotic regions. The range of $\delta$ values is due to the spatial variation of the most contracting FTLE, $\lambda_3$, in the region (see Fig.s 2–3 for most stretching FTLE, which are of the same magnitude). FTLEs were estimated over an integration time of about 20 rotations of the central orbit (dimensionally about three days); the range of FTLE magnitudes does not noticeably change from 10 to 20 rotations. The widths of the chaotic sea and smaller resonant regions were estimated from inspection of Poincaré sections. The Batchelor scale is generally about 0.01–0.08 in nondimensional length units, or about 1–8m dimensionally for $E = 0.25$ and 20–140m for $E = 0.0005$, which are similar to the resonant layer widths and smaller than the chaotic sea widths. The dimensional diffusivities at these scales range from $2 \cdot 10^{-4}$ m$^2$s$^{-1}$ at 1m to 0.06 m$^2$s$^{-1}$ at 140m, which are considerably smaller than diffusivities on the horizontal scale of eddies themselves, about 0.5–8.2 m$^2$s$^{-1}$ for 1–10km. The Batchelor scale results imply that chaotic advection is expected to influence tracer distribution throughout the system, but dominate only in the wider chaotic sea region.

## 4   Particle Dispersion

In this section, we quantify the relative effects of turbulent diffusion and chaotic advection using the dispersion (or spread) of sets of initially nearby trajectories in the kinematic rotating can model. We consider chaotic advection dominant compared to diffusion when the ensemble spread is greater for



the deterministic perturbation that induces chaos than for the stochastic perturbation that simulates
diffusion. Ensembles of one hundred to three hundred trajectories that begin inside a small sphere
have been examined for their behavior under various perturbations. Other initial conditions, on a
torus or axial circle, give similar results (not shown). The spread of trajectories is measured in terms
of $\Psi$ values, the streamfunction of the background flow given by Eqn. (2). Examining the spread in
$\Psi$ is convenient because it leads to zero spread for particles following the background flow. How-
ever, it is important to note that this interpretation limits the directions of chaotic stretching that are
considered— it is possible for the fastest spreading direction to be along the background streamlines,
which would not be visible in the coordinates chosen.

To simulate diffusion, we add a stochastic velocity perturbation to the background model flow.
The stochastic perturbation is a random flight model created by adding small pseudorandom values
with a Gaussian distribution to the velocity at fixed intervals of time $\Delta t$. The equation governing a
fluid particle trajectory is then:

$$\frac{dx_i}{dt} = U_{bi}(\boldsymbol{x}) + u_i', \tag{23}$$

where $i$ is a direction index, $U_{bi}$ is the background velocity, and $u_i'$ are the stochastic additions. These
velocity additions are uncorrelated and lead to a Gaussian random walk behavior (Zambianchi and
Griffa (1994)). With the described stochastic perturbation, the variance of a set of trajectories will
grow linearly in time, while the standard deviation grows linearly with the square root of time, as ex-
pected for diffusion. The diffusivity, $\kappa$, is computed from the 1D relationship for a Gaussian random
walk: $\kappa = s^2/2\Delta t$, where $s$ is the standard deviation of step size in the random walk. To choose the
level of diffusivity for the stochastic perturbation, we consider the diffusivities near the Batchelor
scale as computed in the previous section. The Okubo diffusivities at the Batchelor scale are in the
range $\kappa \in [10^{-4}, 10^{-2}]$m$^2$s$^{-1}$ across the four Ekman numbers examined, which is nondimensionally
$\kappa \in [10^{-6}, 3 \cdot 10^{-5}]$, using the depths of the domains and a velocity scale of 1ms$^{-1}$ to scale diffu-
sivity. As our primary example, we will discuss the level of diffusivity $\kappa = 10^{-6}$. This diffusivity
requires a certain step size $s$ for the stochastic perturbation, which relates to the distribution of $u'$
by $s = \sigma \Delta t/3$, with $\sigma$ the standard deviation of $u'$, $\Delta t$ the numerical timestep (0.01), and the factor
of 3 due to the details of a fourth–order Runga–Kutta integration (the stochastic velocities are added
only at full timesteps). Together, these give

$$\kappa = \frac{\sigma^2 \Delta t}{18}, \tag{24}$$

and so $\sigma = 0.042$. We will also discuss a smaller stochastic perturbation, $\kappa = 10^{-7}$, $\sigma = 0.013$, and
a larger one, $\kappa = 10^{-5}$, $\sigma = 0.13$. The stochastic perturbation with $\kappa = 10^{-6}$ has kinetic energy (in-
tegrated over the cylinder) about the same as the background flow: $\int (\boldsymbol{u_s})^2 \approx \int (\boldsymbol{U_b^2}) \approx 0.63$. The
perturbation with $\kappa = 10^{-7}$ has kinetic energy about the same as the deterministic perturbation with
$\epsilon = 0.01$, $x_0 = -0.5$, such that $\int (\boldsymbol{u_d'})^2 \approx \int (\boldsymbol{u_s'})^2 \approx 0.075$, where $\boldsymbol{u_d'}$ is the deterministic perturba-
tion velocity and $\boldsymbol{u_s'}$ is stochastic.



We begin with an example for $E = 0.125$ showing the spread of trajectories (measured in terms of the background streamfunction $\Psi$) in the presence of either the deterministic or the stochastic perturbation. Trajectories are started on a small sphere located entirely in the chaotic sea region centered on $(r, z) = (0.1, 0.5)$ (see Fig. 2 for the Poincaré section). For the deterministic perturbation at early times, trajectories oscillate through the background streamfunction because the perturbation

velocities form an azimuthal wave (Fig. 6(a)). The frequency of this oscillation depends on the exact location of the trajectory, so with time, trajectories move out pf phase due to the cumulative effect of their slightly different oscillatory frequencies. It takes a few cycles of overturning to develop noticeable spreading, but then the spread grows quickly.

    For the stochastic perturbation (Fig. 6(b)), trajectories are uncorrelated as they spread across the

background streamfunction. There are no oscillations in time because the perturbation acts separately on each trajectory at each timestep, leading to continuous and monotonic spreading of the ensemble. This spreading is similar to diffusion, but the increase in the range of trajectories does not depend on the gradients of concentration the way a diffusing tracer would. If both perturbations are included (Fig. 6(c)), trajectory ensembles maintain some of their oscillatory behavior but spread

out in a more continuous fashion due to the stochastic perturbation. In this example, and over time scales considered, we conclude that the stochastic perturbation dominates at early times but chaotic spreading takes over at times larger than about 1000. Over an even longer time period, diffusive spreading is expected to overtake chaotic spreading.

    We next compare the spreading of trajectory ensembles in $\Psi$ with a variety of perturbations for

the same initial conditions as in Fig. 6 using the range over time (Fig. 7); results are similar when the variance in $\Psi$ is used for comparison (not shown). Chaotic advection dominates when the spread in $\Psi$ for an ensemble under deterministic perturbation is larger than the spread under stochastic perturbation. The spread from the deterministic perturbation appears exponential for a period of time, as expected, but is limited to the width of the chaotic region in which the ensemble begins (e.g.

red curve in Fig. 7(a)). In contrast, the stochastic perturbation will spread with the square root of time until it reaches the cylinder boundaries (e.g. dark blue curve in Fig. 7(a)). Therefore, the time when the deterministic perturbation has greater spread will be limited to between when exponential growth starts in the deterministic perturbation, which requires sufficient interaction with hyperbolic regions, and when the stochastic perturbation spreads the ensemble to the width of the chaotic region.

In the chaotic sea region (left panels of Fig. 7), ensembles with stochastic perturbations all have their ranges in $\Psi$ grow in a manner similar to the square root of time and the spreading is faster for larger $\kappa$. The ensembles with deterministic, chaos–inducing perturbations experience an initial delay before they begin quickly growing. Once rapid growth sets in, they spread to the width of the chaotic region between times 500 and 3000. Larger deterministic perturbations lead to earlier

and faster spreading, as well as wider chaotic regions. For the weaker deterministic perturbation $\epsilon = 0.01$, there are some time intervals over which chaotic spreading in the chaotic sea dominates





stochastic spreading. These instances occur more readily in the case of the shallower eddy ($E = 0.125$, Fig. 7(a)) and less so for the deeper eddy ($E = 0.02$, Fig. 7(c)). However, larger deterministic perturbations (e.g. $\epsilon = 0.08$) produce chaos that is dominant over longer times, an extreme example being the pink curve in Fig. 7(a).

We can also consider the timescales over which diffusive and advective processes with similar kinetic energy (red and light blue curves in Fig. 7) dominate over each other. As discussed for the winding time, the advective timescale is dimensionally 1–5min, which means each section of length $t = 1000$ is about 2 days. The ensembles released in the chaotic sea show that during the first several hours or day, turbulent diffusion dominates the spread (Fig. 7(a)(c) at $t < 1000$), as chaotic advection does not yet show significant growth. After that, on time scales of about one day we see a period of fast growth due to chaotic advection, which quickly overtakes the slower diffusive spreading. This rapid growth stops when the advective spread reaches the width of the chaotic region, and the diffusive spreading, which is not limited by the chaotic region width, is then able to catch up and exceed chaotic advection. Of course, these processes will be acting at the same time, not separately; the green curves in Fig. 7 are examples when small perturbations of both types are present. In this case, spreading of the ensemble begins immediately, as in simulations with only stochastic perturbation, but then has a time period of pronounced growth and some oscillations, as seen in simulations with only the steady perturbations.

We also examined the behavior of trajectories beginning at $(r, z) = (0.4, 0.5)$, a small distance from the central fixed orbit, within the region containing resonant layers (Fig.s 2–3). In these cases, the same behavior as in the chaotic sea region occurs for the spreading under stochastic perturbations (Fig. 7(b)(d)). The spreading under deterministic perturbations is much slower than in the outer chaotic sea region for $\epsilon = 0.01$ (red curves in Fig.s 7(b)(d)) and diffusion dominates at all times for all values of $\kappa$ shown. With $\epsilon = 0.08$, the chaotic region is larger and growth due to the deterministic perturbation is generally more rapid than that due to diffusion, at least within the time window when chaotic advection begins and until saturation occurs (pink curves in Fig.s 7(b)(d)).

From the spreading of ensembles of trajectories, we see that the wider chaotic regions are where chaotic advection dominates over turbulent diffusion (at least over some time intervals), as expected from our scaling arguments. However, those scalings did not include considerations of time, including considerations of when exponential stretching begins; the delay in chaotic stretching decreases the period of time when chaotic advection is important. This time period begins when exponential stretching is first apparent and ends when turbulent diffusion has spread across the region under consideration. From these ensembles, we would expect a set of passive 3D drifters or an injected tracer beginning in a blob to spread out diffusively, then be stretched and folded throughout the chaotic sea, producing strong filamentation, then gradually diffuse across the barriers of the chaotic sea and into the remainder of the eddy. During the later stage tracer variance due to the formation of filaments by



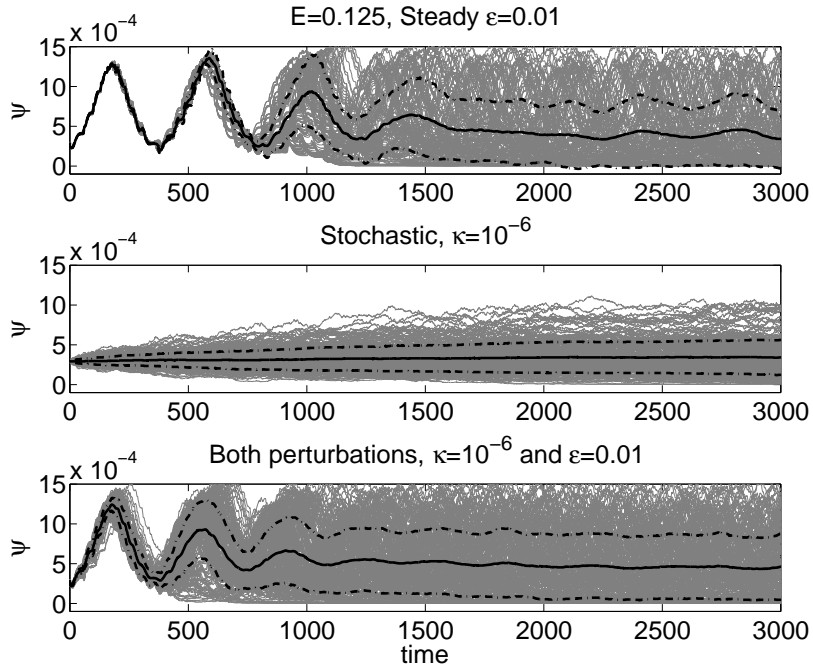

**Figure 6.** Grey lines are individual trajectories in $\psi$ starting from a sphere of radius 0.002 at $(r, z) = (0.1, 0.5)$ with $E = 0.125$. Solid black curves are the mean; black dash–dot lines are $\pm 1$ standard deviations from the mean. Nondimensional time is shown; $t = 1000$ is between 1 and 2 days.

chaotic advection would be gradually eroded by turbulent diffusion. This sequence of events will be apparent in tracer simulations shown in the next section.

## 5 Tracer Simulations and Nakamura Effective Diffusivity


In this section we analyze the effects of the symmetry–breaking, chaos–inducing deterministic velocity perturbation on the stirring and mixing of a diffusive tracer in a dynamically consistent numerical model of a rotating cylinder flow. Dye experiments are often used in both the ocean and the laboratory to understand the stirring and mixing in a fluid (examples include Fountain et al. (2000);

Ledwell et al. (1993, 1998)). The distributions of passive tracers like dye are created by the advective and diffusive patterns without the feedback onto the flow that would occur with temperature or salinity, allowing for insight into those processes. For our simulations we turn away from the kinematic model and take advantage of the existing numerical model that solves Navier–Stokes equations corresponding to the rotating cylinder flow accompanied by integration of the advection/diffusion



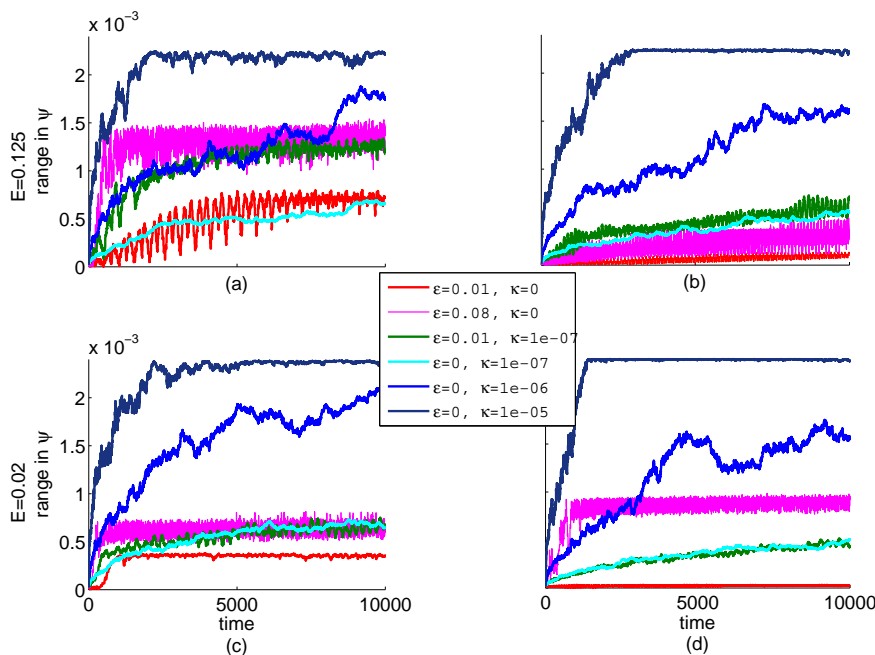

**Figure 7.** Range in $\psi$ for ensembles of trajectories started from a sphere of radius 0.002. Steady perturbation ($\epsilon \in \{0.01, 0.08\}$), stochastic perturbations ($\kappa \in \{10^{-5}, 10^{-6}, 10^{-7}\}$), or both ($\kappa = 10^{-7}, \epsilon = 0.01$), are added to the background flow. Left: Initial sphere in the chaotic sea region, away from fixed points, at $(r, z) = (0.1, 0.5)$. Right: Initial sphere centered on $(r, z) = (0.4, 0.5)$, a resonant region.

equation with diffusivity $k$ for a passive tracer, both described in P2014. As discussed earlier, these simulations have the advantage of being dynamically consistent at the cost of being computationally expensive, whereas economy of the kinematic model allows us to explore a wider range of parameters.

    Our main quantification tool is Nakamura's effective diffusivity: a background diffusivity scaled

by a representation of the stretching of dye concentration contours by advection. Two–dimensional and quasi–three–dimensional analyses of effective diffusivity have been applied to the atmosphere and ocean (Nakamura (1996); Nakamura and Ma (1997); Haynes and Shuckburgh (2000); Abernathey et al. (2010)). For our fully three–dimensional system with constant density, the effective diffusivity can be written as

$$\kappa_{eff}(C) = k \frac{1}{(\partial C/\partial V)^2} \widehat{|\widehat{\nabla C}|}^2, \tag{25}$$

where $C$ is tracer concentration, $V$ is volume, and $\hat{f}$ indicates an average of function $f$ over the area of a concentration surface. The imposed small–scale diffusivity $k$ is constant and so is more closely



related to the $\kappa$ used in Sect. 4 for the stochastic perturbation than the scale–dependent Okubo $\kappa$ in Sect. 3. (It is not clear how one would incorporate a scale–dependent diffusivity into Nakamura's

formulation.) The volume $V$ is a one–to–one mapping of tracer concentration and volume such that $V(C)$ is the volume occupied by fluid with concentrations greater than $C$. The derivation leading to the above definition for $\kappa_{eff}$ can be found in Shuckburgh and Haynes (2003), who perform the algebra in 2D but note that the 3D development is identical. Equation 25 describes an effective diffusivity that is amplified from the small–scale diffusivity by a factor of the degree of contortion

of the concentration contour. The units of the effective diffusivity are those of $k$, typically m$^2$s$^{-1}$, multiplied by m$^4$, or volume squared divided by length squared, which is the same as surface area squared. Larger effective diffusivity leads to larger diffusive fluxes of tracer. This amplification can be understood as being caused by advective stretching and folding of tracer contours which increases the area of surfaces of constant $C$, thereby amplifying gradients of $C$ and speeding up diffusive

fluxes. This amplification factor is precisely the surface area squared in the rare situation where $|\nabla C|$ is constant on a $C$ surface (see Appendix C for proof).

Both advection and diffusion redistribute tracer concentration and influence effective diffusivity. The effective diffusivity allows the effects of advection to be included in a diffusive term:

$$\frac{\partial C}{\partial t} = \frac{\partial}{\partial V}\left(\kappa_{eff}\frac{\partial C}{\partial V}\right). \qquad (26)$$

As advection stretches and folds the initial tracer, creating filaments, the surface area of a contour and gradients of the tracer increase, leading to larger $\kappa_{eff}$. Then, as diffusion smooths the tracer field, wiping away the filaments, gradients decrease and contours become smoother, with a lower surface area to volume ratio. We compare the effective diffusivity with a deterministic perturbation to that without; any increase is due to increased stirring, which gives a quantitative measure of how

important that stirring is for the distribution of tracer in each region of the flow.

As a secondary quantification tool, we use the volume–integrated tracer variance function, $\chi^2$ (Pattanayak (2001)):

$$\chi^2 = \int_V |\nabla C|^2 dV \bigg/ \int_V |C|^2 dV, \qquad (27)$$

where $V$ here is simple volume. Stirring increases the variance of a tracer, while mixing decreases

it. When $\chi^2$ is increasing, stirring is dominant and the slope of $\chi^2(t)$ quantifies the stirring rate. The tracer variance function was used to relate Ekman number, perturbation strength, and stirring rate for the rotating cylinder in P2014; the authors found that stirring increased with larger perturbations and was nonmonotonic with $E$, peaking near $E = 0.01$.

The numerical simulations are run using NEK5000 for several diffusivities and strengths of the

symmetry breaking deterministic perturbation. This model solves the incompressible Navier–Stokes equations using a spectral element method (see https://nek5000.mcs.anl.gov, P2014, Fischer (1997)). The domain has identical radius and height, matching the aspect ratio assumed in our kinematic





model. The symmetry–breaking perturbation is created by moving the central axis of the imposed surface lid stress a distance $X_0$ from the cylinder axis, so that $X_0$ becomes the primary parameter determining the perturbation strength. The $X_0 = -0.02$ case is what was used to compare Poincare sections with the kinematic model, so qualitative features match the $\epsilon = 0.01$ cases. The $X_0 = -0.16$ case is a significantly larger perturbation, similar to the $\epsilon = 0.08$ case in the previous section. The nondimensional imposed tracer diffusivity, $k$, is $10^{-4}$ or $10^{-6}$. Using Okubo's scaling, the lower diffusivity is appropriate for scales near 1m, while the larger is appropriate for scales near 50m. After the simulated velocity field is spun up, the tracer concentration, $C$, is initialized with a constant vertical gradient, $C = 1 - z$.

The set of simulations performed allows for an examination of the effects of changing $E$, $k$, and $X_0$. They are $E = 0.125$, $k = 10^{-4}$, $X_0 \in \{0, -0.02, -0.16\}$ and $E = 0.02$, $k \in \{10^{-4}, 10^{-6}\}$, $X_0 \in \{0, -0.02\}$, for a total of seven simulations. Each simulation is run for a time of 300 after the tracer is initialized. The evolution in time of the tracer variance function and Nakamura effective diffusivity integrated over the volume of the cylinder are described first; we then discuss the evolution of the dye, and finally the spatial characteristics of the Nakamura effective diffusivity.

The tracer variance function over time, Fig. 8, initially grows nearly linearly as stirring creates filaments and large gradients. The function then has a single maximum that occurs at the time when diffusive mixing starts to overcome stirring, so that the variance of the tracer begins to decrease. The maximum occurs earlier when either the imposed diffusivity or the strength of the deterministic perturbation increase. Increasing the diffusivity makes the maximum occur earlier by increasing the strength of the mixing (Fig. 8 (a) to (b)). Increasing the deterministic perturbation also makes the maximum occur earlier as faster stirring creates larger gradients, in turn increasing diffusive fluxes (Fig. 8 (c), red curve).

The maximum of the tracer variance function increases with decreased diffusivity, as more filamentation can occur before diffusion wipes the filaments out. This change of maximum is most evident in the difference between $k = 10^{-4}$ and $k = 10^{-6}$ for $E = 0.02$, where the decrease in diffusivity increases the maximum of the tracer variance function by an order of magnitude (Fig. 8 (a) to (b)). Changes in the maximum as the size of $X_0$ is increased from 0 to 0.02 are small and negative, because the slightly earlier time of maximum combined with similar stirring rates leads to a slightly smaller maximum with the perturbation. In the case of $E = 0.125$, $X_0 = -0.16$, the maximum is larger than with either $X_0 = 0$ or $X_0 = -0.02$ due to faster stirring and a different spatial pattern of the dye, which will be discussed later.

The effective diffusivity, $\kappa_{eff}$, integrated over the total volume shows an overall progression similar to the tracer variance function, which indicates the dominance of the gradient term over both the $\partial C / \partial V$ term in $\kappa_{eff}$ and the $|C|^2$ term in $\chi^2$ (Fig. 8(d)(f)). The initial slope and details of the maximum can be understood as relating to perturbation and diffusivity strengths in the same manner as for $\chi^2$. At longer times, the integrated effective diffusivity reaches a nearly constant positive value



unlike $\chi^2$, which aproaches zero. This constant value can be estimated by using the surface area representation of $\kappa_{eff}$. At long times, here meaning after many overturns but before diffusion removes all gradients, the shape of tracer surfaces are distorted nested tori (see Fig. 9(h)). If the $C$ surfaces were nested circular tori, $|\nabla C|$ would be constant along the surfaces, and then $\kappa_{eff} = kA^2$, where $A$ is the surface area of a given toroidal tracer contour. The volume integral of the squared surface area

of circular tori nested around $(r, z) = (0.5, 0.5)$ multiplied by the background diffusivity is $k\pi^6/8$, which we expect to be the minimum for $\int \kappa_{eff} dV$ in this system while gradients are nonzero (see Appendix C for details). This value is shown as black dashed lines in Fig. 8(d)(e)(f) and is just below the lowest $\int \kappa_{eff} dV$ value seen. The higher values for $\kappa_{eff}$ with steady perturbations at long times corresponds to persistent asymmetries in the tracer field which result in larger constant concentra-

tion surface areas. The extreme case is $E = 0.125$, $X_0 = -0.16$, which has the most asymmetric dye contours; here, the long time value of $\int \kappa_{eff} dV$ is about twice as large as for circular tori.

Further insight can be gained by perusal of vertical sections of $C$ and $\kappa_{eff}$ (Fig.s 9 and 10). A caveat is that $\kappa_{eff}$ is a nonlocal property, the value of which at any point in space and time is influenced by processes occurring at all other locations having the same $C$. Nevertheless, these plots

can yield some insights into the time–histories shown in Fig. 8. Figure 9 is restricted to cases with $E = 0.02$ while Fig. 10 is restricted to $E = 0.125$. The two are laid out differently, with the former designed to emphasize the effects of varying $k$ and the latter designed to explore variations in the strength $X_0$ of the perturbation. Both figures contain snapshots from an early time ($t = 39$) in the simulation, before diffusion has had a chance to arrest growth in the tracer variance function, and at

a late time ($t = 299$) when $\kappa_{eff}$ has reached a quasi–steady value.

The early development ($t = 39$) of the tracer field, $C$, and of $\kappa_{eff}$ can be seen in Fig. 9 (a–f). With no disturbance present ($X_0 = 0$) and $k = 10^{-4}$ (Fig. 9b), the initially horizontal lines of constant $C$ have been advected by the axially symmetric overturning circulation such that contours of constant $C$ are roughly aligned with the overturning streamfunction. The corresponding $\kappa_{eff}$ (Fig. 9(e)) exhibits

high values at the edges of filaments created by the straining motion of the symmetric background flow, despite the fact that no trajectories are chaotic. When a disturbance is added ($X_0 = -0.02$, Fig.s 9(c)(f)) the axial symmetry is broken and the peak values of $\kappa_{eff}$ are reduced. The latter is somewhat surprising since we have already seen (Fig. 8(b)) that the volume integrated values of $\kappa_{eff}$ are nearly the same for the disturbed and undisturbed case. The situation is made clearer if one

notes that moderate values of $\kappa_{eff}$ (yellow in Fig. 9(f)) are more widely distributed in the disturbed case. A similar result can be seen by comparing the case $X_0 = 0$ (Fig. 10(a)(d)) to $X_0 = -0.02$ (Fig.s 10(b)(e)), all for E=0.125. Again, the unperturbed (symmetric case) has larger peak values while the perturbed case has more locations with moderate values of $\kappa_{eff}$, resulting in a similar volume integrated value of $\kappa_{eff}$ (Fig. 8(f)). It is possible that slight increases in stirring in the

perturbed cases has caused more mixing than in the unperturbed cases, even over the short interval before these snapshots, leading to a lower range of $C$ and smaller average gradients in the perturbed





cases. However, the volume–integrated measures (Fig. 8) do not show any clear indications of that process occurring.

When the background eddy diffusivity $k$ is decreased by two orders of magnitude, with $X_0$ fixed at

$-0.02$, the results are remarkably different. To begin with, a comparison of Fig. 8(d) with 8(e) shows that $\kappa_{eff}$ is generally larger at any particular time when $k$ takes the smaller value. As Fig. 9(a) and (c) show, the tracer field contains much finer filaments when $k = 10^{-6}$, consistent with the reduction of the Batchelor scale. The distribution of $\kappa_{eff}$ is broader and with larger peak values for this lower numerical diffusivity (compare Fig. 9(d) and (f)). The higher $\kappa_{eff}$ indicates that despite the decrease

in $k$, the effects of stirring on the contours, $\kappa_{eff}/k$, have more than compensated, resulting in a higher rate of irreversible property exchange. Thus the combined effect of smaller diffusivity and finer filaments (i.e., stronger tracer gradients) leads to more rapid mixing across tracer contours.

The results that have just been described occur early ($t = 39$) in the evolution of the tracer field, at a time when fluid parcels have overturned just a few times and the perturbation amplitude $X_0$

has been small. For this weakly perturbed flow, Lagrangian chaos requires many overturns to be significant, so we now turn attention to the results for $t = 299$ (bottom six panels of Fig.s 9 and 10). Here a comparison between the unperturbed and perturbed cases (contrast panels 9(h)(k) with 9(i)(l) and also 10(g)(j) with 10(h)(k)) reveal only modest differences in the spatial distribution and magnitude of $C$ and $\kappa_{eff}$. As in the early snapshots, there is a tendency for the unperturbed flows

to have higher peak values of $\kappa_{eff}$, while the perturbed flows produce moderate values over a larger area. Decreasing the value of $k$ again has the effect of creating more fine structure (Fig. 9(g)) and of increasing the peak values of $\kappa_{eff}$ by an order of magnitude (Fig. 9(j)).

So far, the consequences of the symmetry–breaking disturbance are modest. However, dramatic differences occur when $X_0$ is increased from $-0.02$ to $-0.16$ for $E = 0.125$ (right–hand panels

in Fig. 10). The tracer distribution is markedly distorted at early times (compare 10b with 10c) and strong tracer gradients remain present even at $t = 299$, at a time when the gradients in the unperturbed and weakly perturbed cases have been strongly eroded (compare 10(g)(h) with 10(i)). The peak values of $\kappa_{eff}$ at $t = 299$ (Fig. 10(l)) remain comparable to those of the weakly perturbed case (10(k)) but occupy a much larger volume, making the volume integrated $\kappa_{eff}$ much larger, in

agreement with Fig. 8(f).

For a different perspective, we examine the mean $\kappa_{eff}$ in subdomains of the system corresponding to a regular island and a region of the chaotic resonant layer of roughly the same size. The cross–sections of the cylinder along the $x$ and $y$ axes are broken into different regions using the matching Poincaré sections of the perturbed flow (Fig. 11). Demarcation of these subdomains was

most straightforward for the case $E = 0.02$, due to its large island and extended resonant region. The mean $\kappa_{eff}$ in the chosen subdomains gives a clear result in the $E = 0.02$, $k = 10^{-4}$ case (Fig. 11(c)), where at long times, when the overall gradients have smoothed out, the resonant regions have about twice the effective diffusivity as the islands. The islands' $\kappa_{eff}$ at that time approximately

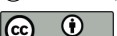



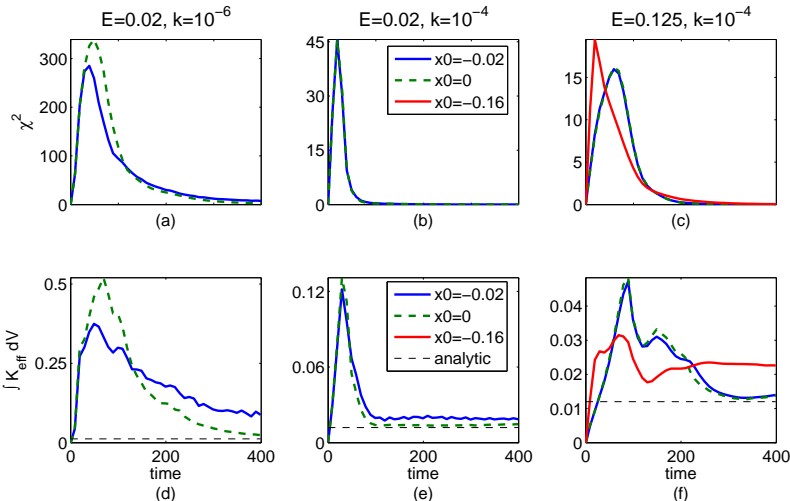

**Figure 8.** Top, tracer variance, $\chi^2$; bottom, $\kappa_{eff}$ integrated over volume. Left: $k = 10^{-6}$, $E = 0.02$, middle: $k = 10^{-4}$, $E = 0.02$, right: $k = 10^{-4}$, $E = 0.125$. Solid blue lines include the deterministic perturbation which induces chaos, $\epsilon = -0.02$, green dashed lines are unperturbed, solid red lines include the deterministic perturbation with $\epsilon = -0.16$. Black dashed lines indicate $\kappa_{eff}$ integrated over volume in the case of nested circular tori.

matches the value from the same region in the unperturbed simulation, indicating that chaos has not

affected this area. In the $E = 0.02$, $k = 10^{-6}$ case (Fig. 11(d)) the mean $\kappa_{eff}$ is typically higher in the resonant region than in the island, but the differences are less pronounced. It is notable that at $t > 130$, $\kappa_{eff}$ is larger in the island than in the same unperturbed region, perhaps because islands are not completely regular and contain smaller chaotic resonant regions within them.

Overall, these dye experiments show that chaotic advection enhances Nakamura effective diffu-

sivity within the chaotic sea at some times in all cases examined. The amount of enhancement is controlled by both the size of the perturbation and the imposed diffusivity. A larger perturbation leads to greater enhancement ($\kappa_{eff}$). A smaller diffusivity leads to less mixing ($\chi^2$) and highly elevated enhancement ($\kappa_{eff}$).

## 6   Conclusions

The main thrust of this work is to establish whether the stirring due to chaotic advection in an idealized model of an upper ocean eddy remains relevant in the presence of levels of background turbulence that are consistent with observations. The answer is that chaotic advection can indeed be relevant, and in some cases dominant, within certain regions of the flow field and over certain time intervals. The region most likely to feel the effects of chaotic advection is the extensive chaotic

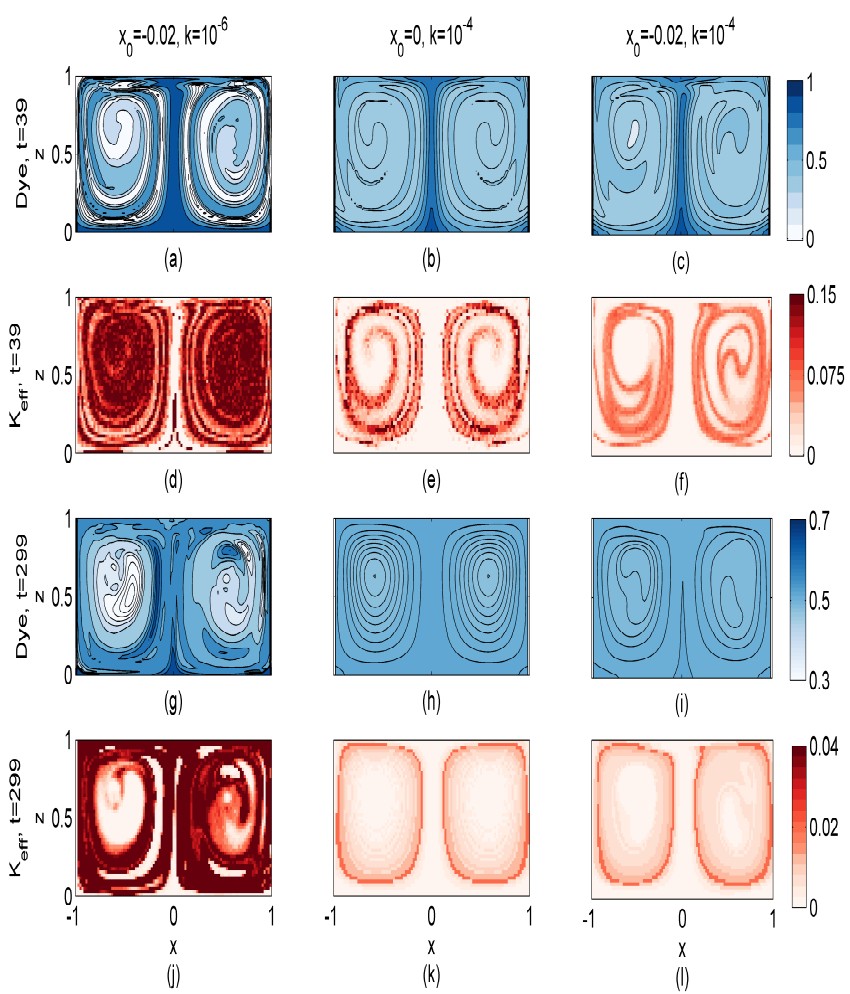

**Figure 9.** Results from three Navier–Stokes simulations with $E = 0.02$: left, $x_0 = -0.02$, $k = 10^{-6}$, middle, $x_0 = 0$, $k = 10^{-4}$, right, $x_0 = -0.02$, $k = 10^{-4}$. The $x_0 = 0$, $k = 10^{-6}$ case is not shown, but is qulitatively similar to the $x_0 = 0$, $k = 10^{-4}$ case. Top: Dye, $t = 39$. Row 2: $\kappa_{eff}$, $t = 39$. Row 3: Dye, $t = 299$. Bottom: $\kappa_{eff}$, $t = 299$.


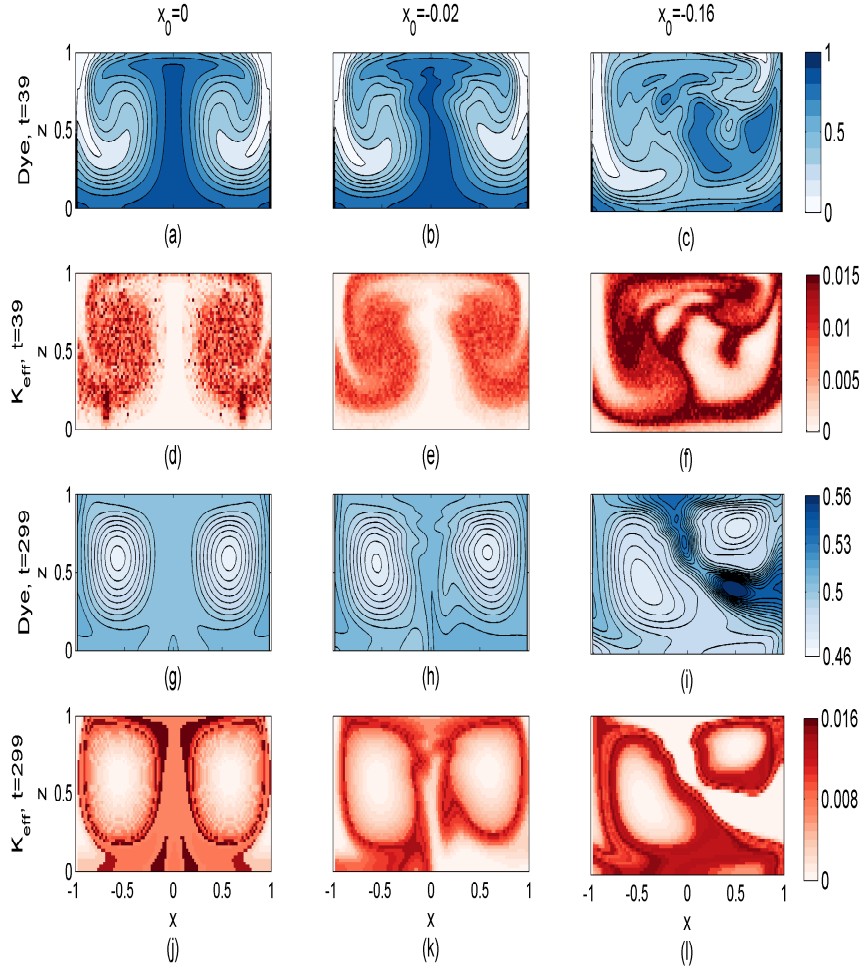

**Figure 10.** Results from Navier–Stokes simulations for $E = 0.125$ with three deterministic perturbation levels: left, $X_0 = 0$; middle, $X_0 = -0.02$; right, $X_0 = -0.16$. Top: Dye, $t = 39$. Row 2: $\kappa_{eff}$, $t = 39$. Row 3: Dye, $t = 299$. Bottom: $\kappa_{eff}$, $t = 299$.





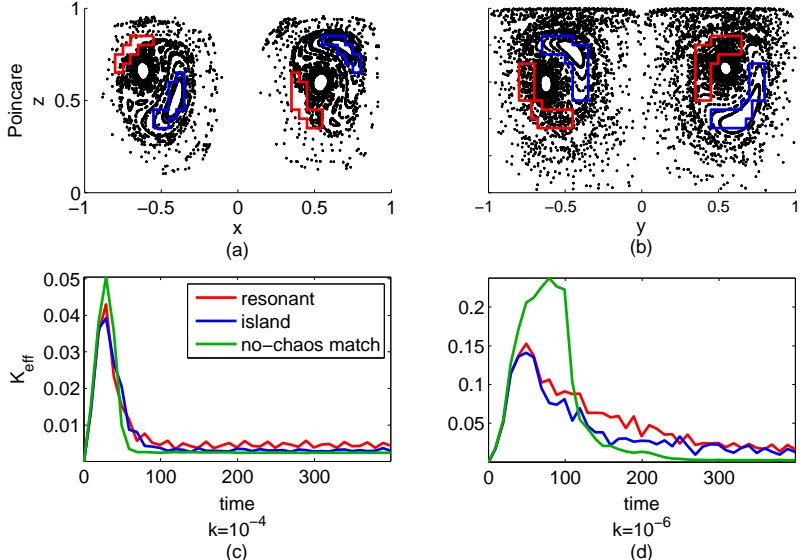

**Figure 11.** $E = 0.02$ Poincaré sections in the (a)$x$–$z$ and (b)$y$–$z$ planes in black. Polygons show the island (blue) and resonant (red) regions used for analysis (c) and (d), mean $\kappa_{eff}$ over time in these regions under both applied background diffusivities.

sea that exists in all simulations, and is especially pronounced when the eddy is shallow. Chaotic stirring in the smaller and more isolated resonant regions is less likely to be important. This conclusion comes with many caveats related to idealizations (e.g. homogeneous turbulence) and uncertain parameter values (e.g. background diffusivity, strength of perturbation).

A second focus of the work has been to explore different bases for comparison of the effects

of chaotic advection and homogeneous turbulence. To this end we have identified three metrics for comparison and are now in a position to discuss their pros and cons. The first metric is the Largrangian Batchelor scale (Sect. 3), an estimate of the equilibrium width of a passive tracer filament. Equilibrium is achieved when transverse compression due to advection, as quantified by the negative Lyapunov exponent with the largest magnitude ($\lambda_3$) is balanced by the diffusive spreading

of the tracer. Below the Batchelor scale, diffusion is stronger than advection; when this width is larger/smaller than that of the chaotic regions, diffusion/advection dominates. We fixed the turbulent diffusivity using Okubo's empirical formula and calculated the Batchelor scale $\delta$ using the rate of chaotic filament stretching, $\lambda_3$, computed numerically as the largest negative finite–time Lyapunov exponent for the kinematic model. The resulting Batchelor scale varies from O(1m) for $E = 0.25$

to O(100m) for $E = 0.0005$. These values of $\delta$ are smaller than the spatial extent of the chaotic sea




over all $E$ values considered (0.25, 0.125, 0.02, and 0.0005), but of similar magnitude to the widths of the resonant regions.

Interpretation of the Lagrangian Batchelor scale analysis would appear to be straightforward, but it does not comprehend the fact that chaotic advection may only be dominant over a finite time
interval. Even when the level of background turbulence is weak, turbulent diffusion will eventually spread beyond the region of Lagrangian chaos. There is also a level of uncertainty due to the choice of integration time over which $\lambda_3$ is calculated. Finally, it is not yet possible to calculate $\lambda_3$ from ocean data with contemporary float/drifter technology. Vertical velocities are typically very weak and Lagrangian drifters that are able to follow water parcels in 3D are expensive and have only been
deployed in small numbers D'Asaro et al. (1996); D'Asaro (2015).

As a second basis for comparison, we computed the dispersion over time of initially small clusters of trajectories (Sect. 4) as they spread across isosurfaces of the background streamfunction. Background turbulence is simulated as a Lagrangian random walk based on spatially uniform diffusivity. We consider the dispersion characteristics that arise when this representation of turbulent diffusion
is added to a background flow with no chaotic advection and compare it to flows that are undergoing chaotic advection but lack turbulent diffusion. Since the chaotic regions occupy sub–volumes of the entire eddy, spread of trajectories or tracers due to turbulent diffusion will eventually surpass that due to chaotic advection: chaos alone cannot distribute parcels across Lagrangian boundaries. However, it remains meaningful to compare the rate of spreading of parcels at earlier times. One immediate
observation is that the character of ensemble spreading is qualitatively different for advective as opposed to diffusive perturbations. For the former, the spreading rate is significantly enhanced at some key times when trajectories pass near strong hyperbolic regions. In the latter case, the spread grows similarly to the square–root of time at all times.

When the eddy is moderately shallow ($E = 0.125$) there are many instances in which chaotic
advection in the chaotic sea dominates turbulent diffusion, even at the higher ranges of turbulent diffusivity. When the perturbation strength is moderately large ($\epsilon = 0.08, x_0 = -.02$), chaotic advection produces more rapid spreading than diffusion for two of three diffusivities considered (pink curve in Fig. 7(a)). Even when the perturbation strength is small ($\epsilon = 0.01$), spread due to chaotic advection in the chaotic sea (red curve in 7(a)) is of comparable order to turbulent diffusion at the lowest $k$
values considered (light blue curve in 7(a)). These results are in agreement with the Batchelor Scale analysis.

When the eddy is deeper (E=0.02) spreading due to turbulent diffusion in the chaotic sea and resonant regions generally dominates over spreading due to chaotic advection. This holds even when the perturbation strength is moderately large ($\epsilon = 0.08$). These results are not in strict agreement with
the Batchelor Scale analysis (Fig. 5) result that the dimension of the chaotic sea is greater or equal to that of the Lagrangian Batchelor scale for deeper eddies. To reconcile these inconsistencies, note that as $E$ gets small, a greater percentage of the eddy volume becomes occupied by an inviscid, vertically





rigid interior. For very small $E$, parcels experience relatively low levels of strain while rising or descending through the region. When a fluid parcel nears the top or bottom boundary, however, it become vertically squashed and horizontally stretched, suggesting that the main contribution to $\lambda_3$ comes from close encounters with these boundaries. A Batchelor scale that is based only on a single parameter measuring the time–averaged contraction over several overturning cycles may be too simplistic when a parcel divides its time between kinematically distinct regions.

This method of comparison based on parcel spreading has several advantages over the Batchelor scale. First, it offers a direct measure of fluid stirring. Also, it reveals information about the time history of dispersion that is hidden in the Lagrangian Batchelor scale analysis. Disadvantages include the fact that the analysis, as presented, does not account for scale–dependent diffusivity. Also, like the Batchelor scale analysis, it requires the tracking of fluid parcels in 3D, something that is currently difficult in the ocean. The third method for comparison (Sect. 5) differs from the first two in that it is based on metrics of irreversible property exchange (mixing). These metrics consist of the Nakamura effective diffusivity, $\kappa_{eff}$, and a volume–integrated tracer variance function, $\chi^2$. We consider a flow with a given background turbulent diffusivity, $k$, and calculate how much the irreversible property exchange is amplified as a result of chaotic stirring. The volume–integrated $\kappa_{eff}$ and $\chi^2$ both depend on time and show rapid initial growth, a result of filamentation of an initially smooth tracer distribution. Growth is arrested when diffusion begins to dominate due to the enhanced gradients produced by the filamentation process, at which time both measures, $\kappa_{eff}$ and $\chi^2$, reach peak values. This is followed by a long period in which $\chi^2$ slowly diminishes to zero and the volume integral of $\kappa_{eff}$ reaches a nearly constant value. In most cases, chaotic advection leads to more rapid initial growth, a lower peak value for both measures, and a larger long–term, near–equilibrium value of $\kappa_{eff}$. In weakly perturbed cases, the differences in initial growth and peak value of $\kappa_{eff}$ are minor, usually on the order of 10 or 20%, while differences in the longer term, near–equilibrium value of $\kappa_{eff}$ are more significant. For strongly perturbed cases the initial growth is an order of magnitude larger and the amplification in the long–term value of $\kappa_{eff}$ is larger by a factor of two than in the unperturbed case.

The spatial structure of $\kappa_{eff}$ also yields interesting information, though one must be aware of the caveat that the local value is due to non–local processes. The chaotic sea region generally has enhanced values compared to the interior and its resonant regions. Under weak perturbation, maximum values of $\kappa_{eff}$ were smaller than in the unperturbed case, but the spatial extent of the intermediate values was larger, leading to the enhanced volume–integrated values discussed above. Larger changes in $\kappa_{eff}$ are evident for lower $k$ due to the occurrence of more numerous small–scale filaments. With a larger perturbation, chaotic advection dramatically changes the effective diffusivity, but there are also stronger barriers present, evident from isolated areas with different tracer concentration. We conclude that the spatial structures of chaotic and regular regions can play an important role in how a tracer is distributed.





The use of effective diffusivity as a metric has several advantages and disadvantages. First of all, it provides a direct measure of irreversible property exchange between regions with different dye concentration. Its time history leads to insights about the evolution of mixing and, in particular, the time periods when chaotic advection is most relevant. Also, it can be measured, at least in principle, by performing an ocean dye release and measuring the dye concentration along sections that cut
through the dye plume at different depths or angles, all in an attempt to recreate a concentration map in 3D. Of the three methods proposed herein, it would appear to be the one most testable by ocean observations. The main disadvantage of effective diffusivity is that it requires the background diffusivity to be constant, which is strictly true only if the diffusivity is interpreted as the molecular diffusivity.

In this work, we examined the relative strengths of advection and diffusion for the redistribution of a passive tracer in a rotating cylinder flow as an analogue for an overturning submesoscale eddy. Since a major challenge of this work has been to develop ways of thinking about the competition between chaotic advection and turbulent diffusion, the numerical experiments described in this paper have been necessarily idealized. Exploration with models that are more realistic for the
ocean presents a number of challenges, including the development of more anisotropic and spatially–varying representations of turbulence to account for differences between the ocean surface mixed layer and the stratified fluid underneath. In addition, finite eddy lifetimes must be confronted as a separation of timescales between feature lifetimes and the periods of trajectories within them is needed for these analyses.



## Appendix A: Bifurcation Analysis of Fixed Points of the Background Streamfunction


Here we provide detail about the fixed points, and their bifurcations, of the background velocity field in the kinematic model of the rotating cylinder. Then we present the bifurcation diagram and an example of the flow with many fixed points in the overturning streamfunction.

The overturning streamfunction is described by Eqn.s (2–7), with radial and vertical velocities (8–9) and azimuthal velocity (10). All 3 velocity components are zero at $z = 0$ and $r = a$. The azimuthal velocity $V$ only has 1 other zero at $r = 0$. However, there exist additional points with zero vertical and radial velocity, which correspond to circular periodic orbits in the horizontal plane and which we refer to as $rz$–fixed points.

All $rz$–fixed points in the interior occur at $r = 0.5a$, because this is the only place where $W = 0$. Finding the $rz$–fixed points is thus equivalent to finding points in $z$ where $U(r = 0.5a, z) = 0$. One such point exists for all $E$ at $z = 0.5$. Additional $rz$–fixed points appear through pitchfork bifurcations, where new pairs split from $z = 0.5$ and move apart in $z$ as $E$ decreases from one (Fig. 12).

It is possible to classify the rz–fixed points as elliptic or hyperbolic according to their behavior in the $r$–$z$ plane: the overturning streamfunction is a local maximum in both $z$ and $r$ at elliptic points and a saddle, i.e., a mininum in $r$ but a maximum in $z$, at hyperbolic points. At $E = 1$, the only stationary point is at $(r, z) = (0.5, 0.5)$, and it is elliptic. As $E$ decreases to about $1/62$, the first bifurcation creates two elliptic points above and below the now–hyperbolic central point at $(r, z) = (0.5, 0.5)$. As $E$ decreases, the newly created points move away vertically from the central point, until the next bifurcation creates two new hyperbolic points, and the central fixed point becomes elliptic again. This process continues; the number of fixed points increases as $E$ decreases through a repeated pitchfork bifurcations of the $(r, z) = (0.5, 0.5)$ fixed point. As these bifurcations occur, their effects remain within a region bounded by trajectories between the first pair of hyperbolic points, meaning that their effects are quite local. The spreading of the first pair of hyperbolic points, and not the total increase in rz–fixed points, causes the increasing vertical homogeneity of the flow with decreasing $E$ which appears qualitatively similar to Taylor columns. An example with 9 $rz$–fixed points is shown in Fig. 13 for $E = 0.00125$; the central point is now elliptic. Trajectories in the vertical plane are level curves of the streamfunction; these show the elliptic and hyperbolic nature of the $rz$–fixed points, where trajectories near an elliptic point remain nearby but trajectories near a hyperbolic point may travel a long distance before returning or may move toward another hyperbolic point.

## Appendix B: Gaussian Tracer in Linear Strain

In this appendix, we present the derivation of the evolution of a three–dimensional tracer in a steady linear strain flow. This result was used in the main text to show that the thinnest width of the Gaussian




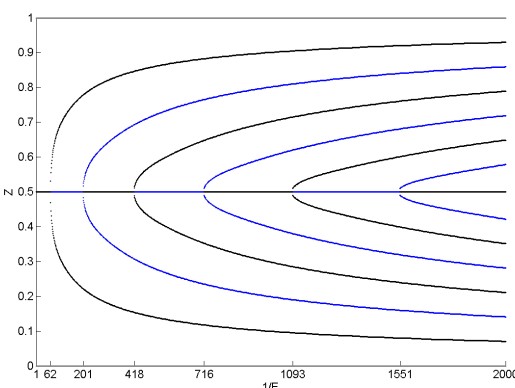

**Figure 12.** $z$–positions of $rz$–fixed points. Black indicates elliptic points, blue hyperbolic, gray the neutrally stable points at the top and bottom. New fixed point pairs separate symmetrically from $z = 0.5$ as $E$ decreases. At each bifurcation, the central fixed point changes stability.

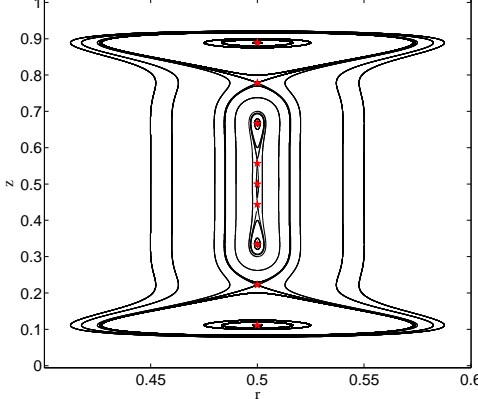

**Figure 13.** Trajectories in the vertical plane for $E = 0.00125$, $a = 1$. There are 9 rz–fixed points along $r = 0.5$, marked with red stars. Note the closed curves between the outermost hyperbolic points which surround the interior 5 rz–fixed points; these limit the effects of those points to the local area.




tracer distribution will asymptotically approach the Lagrangian Batchelor scale. We start with the definitions of the velocity field, the tracer evolution equation, and the form of the solution. Then we derive the full time–dependent solution for the tracer distribution.

We are solving for the evolution of tracer concentration, $C$, with a solution in the form of a Gaussian function

$$C = c_{max}(t) \exp\left( \frac{-x^2\alpha^2(t)}{2} + \frac{-y^2\beta^2(t)}{2} + \frac{-z^2\gamma^2(t)}{2} \right), \tag{B1}$$

where $c_{max}$ is the maximum concentration and $\alpha$, $\beta$, $\gamma$ are the reciprocal of the standard deviations in each direction. In the Lagrangian frame of reference that is moving with the center of mass of the tracer, these four parameters are dependent on time but not space. The smallest width of the distribution is $\sigma = 1/\alpha$ and in the main text we have used the fact that it has a stable fixed point

$\sigma = \sqrt{\kappa/|\lambda_3|}$, where $\lambda_3$ is the contraction rate of the velocity field. We are now going to formally prove it.

The velocities are defined in the Lagrangian frame by

$$u = \lambda_3 x(\boldsymbol{x_0}, t), \tag{B2}$$

$$v = \lambda_2 y(\boldsymbol{x_0}, t), \tag{B3}$$

$$w = \lambda_1 z(\boldsymbol{x_0}, t), \tag{B4}$$

$$\lambda_1 > \lambda_2 > \lambda_3, \tag{B5}$$

$$\lambda_1 > 0, \ \lambda_3 < 0, \tag{B6}$$

with $\boldsymbol{x}(\boldsymbol{x_0}, t)$ indicating the initial position $\boldsymbol{x_0}$ of the water parcel at $t = 0$. The Lagrangian tracer evolution equation is

$$\frac{\partial C}{\partial t} + \lambda_3 x \frac{\partial C}{\partial x} + \lambda_2 y \frac{\partial C}{\partial y} + \lambda_1 z \frac{\partial C}{\partial z} = \kappa \nabla^2 C, \tag{B7}$$

where $\kappa$ is the diffusivity.

The form of $C$ and the tracer evolution equation allow us to find differential equations for each of our four parameters, which are

$$\frac{1}{c_{max}} \frac{dc_{max}}{dt} = -\kappa \left( \alpha^2 + \beta^2 + \gamma^2 \right), \tag{B8}$$

$$\frac{d\alpha}{dt} = -\lambda_3 \alpha - \kappa \alpha^3, \tag{B9}$$

$$\frac{d\beta}{dt} = -\lambda_2 \beta - \kappa \beta^3, \tag{B10}$$

$$\frac{d\gamma}{dt} = -\lambda_1 \gamma - \kappa \gamma^3. \tag{B11}$$



The width parameters' equations are nonlinear, but rewritten in terms like $\alpha^{--2}$ give:

$$\frac{d\alpha^{-2}}{dt} = 2\lambda_3\alpha^{-2} + 2\kappa, \tag{B12}$$

$$\frac{d\beta^{-2}}{dt} = 2\lambda_2\beta^{-2} + 2\kappa, \tag{B13}$$

$$\frac{d\gamma^{-2}}{dt} = 2\lambda_1\gamma^{-2} + 2\kappa, \tag{B14}$$

which are Bernoulli equations, solvable with integrating factors, giving

$$\alpha = \sqrt{|\lambda_3|/\kappa}\left((\lambda_3\alpha_0^{-2}/\kappa - 1)e^{2\lambda_3 t} + 1\right)^{-1/2}, \tag{B15}$$

$$\beta = \left((\beta_0^{-2} + \kappa/\lambda_2)e^{2\lambda_2 t} - \kappa/\lambda_2\right)^{-1/2}, \tag{B16}$$

$$\gamma = \sqrt{\lambda_1/\kappa}\left((\lambda_1\gamma_0^{-2}/\kappa + 1)e^{2\lambda_1 t} - 1\right)^{-1/2}, \tag{B17}$$

where subscript $0$ indicates the value at $t = 0$. The differences in these equations is due to the different signs of each $\lambda$, with the ambiguity of the sign of $\lambda_2$ preventing its factoring.

The $c_{max}$ equation depends on the width parameters and is not simple to solve directly. However, a careful inspection shows that $c_{max}/(\alpha\beta\gamma)$ is conserved, so we can write

$$c_{max}(t) = c_0\alpha(t)\beta(t)\gamma(t). \tag{B18}$$

For anyone in doubt, we plug in this solution to check it:

$$\frac{dc_{max}}{dt} = \frac{d}{dt}(c_0\alpha\beta\gamma) = c_0\left(\beta\gamma\frac{d\alpha}{dt} + \alpha\gamma\frac{d\beta}{dt} + \alpha\beta\frac{d\gamma}{dt}\right),$$

$$= c_0\left(-\alpha\beta\gamma(\lambda_3 + \kappa\alpha^2) - \alpha\beta\gamma(\lambda_2 + \kappa\beta^2) - \alpha\beta\gamma(\lambda_1 + \kappa\gamma^2)\right),$$

$$= -c_0\alpha\beta\gamma\left(\lambda_1 + \lambda_2 + \lambda_3 + \kappa[\alpha^2 + \beta^2 + \gamma^2]\right),$$

$$\implies \frac{1}{c_{max}}\frac{dc_{max}}{dt} = -\kappa\left(\alpha^2 + \beta^2 + \gamma^2\right).$$

The full solution for the tracer concentration $C$ then has been fully solved by (B1) with $\alpha, \beta, \gamma$ and $c_{max}$ given by (B15–18).

For a three dimensional Gaussian tracer advected by a linear strain field in the presence of constant diffusivity, in the Lagrangian frame the width of the tracer distribution will increase in the stretching
direction(s) forever, but reach a fixed value in the contracting direction(s).

## Appendix C: Long–Time Limit of Effective Diffusivity For The Axially–Symmetric Rotating Cylinder Flow

For the axially–symmetric rotating cylinder flow at long times, the dye contours resemble nested tori, although with cross–sections that are somewhat between a circle and a square. Here, we derive
the expected limit of $\int \kappa_{eff}dV$ assuming that the dye iso–contours at late times are nested tori with a circular cross–section, and that the gradient of the dye concentration is constant along each torus. In


this case the effective diffusivity on each torus is $\kappa_{eff} = kA^2$, the background diffusivity multiplied by the squared surface area of a torus.

Recall that the volume of a circular torus is

$$V_{ct} = 2\pi^2 r^2 R, \tag{C1}$$

where $r$ is the radius of the circular cross–section and $R$ is the distance from the center of mass of the torus to the center of the cross–section. The surface area is

$$A_{ct} = 4\pi^2 rR.$$

Noting that $A_{ct} = dV_{ct}/dr$, we can calculate the volume–integrated effective diffusivity as

$$\int \kappa_{eff} dV = \iiint kA^2 dV$$
$$= k \int_0^{r_{max}} A^3 dr$$
$$= k \int_0^{r_{max}} (4\pi^2 rR)^3 dr$$
$$= 4^3 \pi^6 R^3 k \int_0^{r_{max}} r^3 dr$$
$$= 4^2 \pi^6 R^3 kr^4 \Big|_0^{r_{max}} = k\pi^6/8 \tag{C2}$$

using $R = 0.5$ and $r_{max} = 0.5$. This circular–torus–based result gives a lower bound, because there is still volume outside the largest torus that fits in the cylinder and the final cross–sections are some-what square, thus having a larger surface area per volume.

*Acknowledgements.* This work was supported on DOD (MURI) Grant No. N000141110087 as well as US Na-tional Science Foundation Grant OCE–1558806. G. Brett received additional support from the Woods Hole
Oceanographic Institution Academic Programs Office. We also thank Marianna Linz and Dan Collura for help-ful conversations on how to present this material.

*Code availability.* Matlab codes for performing trajectory integrations and for making most figures are available at Github (Brett (2018)). Data (code outputs) for making the figures is availabe through Zenodo (Brett and Wang (2018)). Outputs of NS simulations using NEK5000 are included, but the
source code is available instead at *https://nek5000.mcs.anl.gov*.

*Author contributions.* .

G. Brett carried out the main investigation and formal analyses of this work and wrote the initial draft. L. Pratt and I. Rypina aquired funding, formulated the kinematic model, and supervised the project. P. Wang performed



the NS simulations and contributed to their design and analysis. Pratt, Rypina, and Wang revised the written

work, with L. Pratt contributing significant portions of the tfinal text.

*Competing interests.* The authors declare that they have no conflicts of interest.





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
