# Peer review of "Competition between Chaotic Advection and Diffusion: Stirring and Mixing in a 3D Eddy Model"

_Nonlinear Processes in Geophysics, 2018_

## Referee Comment (RC1) · Anonymous Referee #1 · 2 Jan 2019

In this article, the authors present results of an extensive series of studies about the relative importance of chaotic advection (versus turbulent diffusivity) in the dispersion of passive tracers in a model of an ocean-scale vortex flow. The flow itself is a "rotating can" (or rotating cylinder) flow with an optional perturbation that approximates the effects of a differentially-rotating lid that is slightly off-center – this perturbation is responsible for the chaotic behavior of the advection in this flow (there would be no chaos if the flow were perfectly axisymmetric). The paper includes results from both a kinematic (phenomenological) model of this flow and a flow based on numerical solutions to Navier-Stokes equations. The authors present three different analyses of transport in this flow.

1. The first is based on a very simple (and illuminating) approach in which the "Batchelor" length scale (the scale at which thinning of tendrils from chaotic advection is balanced by diffusive smearing) is compared to the size of the chaotic region. This is a very nice approach that can very quickly give good insight into the relevance of chaotic advection in ocean-scale flows, with chaotic advection being important in cases where the chaotic region is wider than the Batchelor scale.

2. A second approach involves the direct simulation of the spreading of an ensemble of tracers in the flow, with added perturbations. The authors distinguish two types of perturbations: either stochastic noise which models the effects of turbulent diffusivity and/or "deterministic" perturbations (the asymmetry) that gives rise to chaotic advection. The simulations demonstrate the relative importance of these two effects.

3. The third approach involves a measure called "Nakumura effective diffusivity" that accounts for both stochastic diffusivity and also the effects of advection which are incorporated into the measure by characterizing the degree of stretching and folding in the flow. This third section also considers the variance $\Chi^2$ that grows as advection makes thinner and thinner filamentary structures but which then gets smaller as diffusion smears those structures away. The authors present detailed simulations both of the spatially-averaged Nakumura diffusivity (for different values of added noise and non-symmetric perturbations), along with spatial maps of the local diffusivity and the concentration profiles for evolving tracer distributions.

The results of all of these studies is a very well-considered and demonstrated conclusion that chaotic advection can be very important in understanding ocean dispersion, although it depends critically on the size of the chaotic mixing regions in the flow.

This is an excellent paper, and clearly deserves to be published in NPG. This is a very important issue; i.e., how relevant chaotic advection is to understanding mixing in real oceanic flows,and the arguments made and the analytical techniques discussed are very compelling. The presentation is clear and very comprehensive – the authors have left nothing out. In fact, there is so much detail that other investigators could replicate

the results with the information provided here without any need for other references. This will be a good reference for others wishing to look at chaotic mixing in oceanic flows.

I have a couple of small comments that the authors may wish to consider if revising the paper. These are all optional – the article is fine if the authors choose to publish as is.

1. A sketch of the flow would be helpful when first discussed in detail, showing both the rotating flow, the Ekman layers, and the off-center perturbation.

2. The Batchelor scale is a little difficult to follow when first presented at the beginning of section 3, but then there is an outstanding discussion of this in section 3.1 (on p. 11) with ample references. Perhaps move that discussion a little earlier.

3. I will admit that by the time I got halfway through section 5, I was beginning to fade. Although I think that the detail is useful overall, perhaps some trimming of section 5 would be helpful to the readability of the paper. But this isn't critical – it's fine if the authors leave it as is.

---

## Referee Comment (RC2) · Anonymous Referee #2 · 4 Jan 2019

Review: Competition between Chaotic Advection and Diffusion: Stirring and Mixing in a 3D Eddy Model

The authors study the transport of a passive tracer in a model of an oceanic eddy with regard to the effect of a deterministic disturbance and an additional diffusivity. The oceanic eddy is modeled by two approaches, first a kinematic model is elaborated upon and then compared to a DNS simulation. Both models share the idea of a rotating cylinder with an additional stirring at the top by a differentially rotating lid. The deterministic disturbance is introduced by a displacement of the axis of rotation. Both models give qualitatively similar chaotic states which are analyzed by Poincare maps. Further different approaches are taken in order to introduce some diffusive process to the system. These approaches cannot be directly compared since they are

implemented within the different eddy models but qualitatively the results hint into the same direction. Especially interesting is the variety and detailed analysis of three different measures that determine the transport, mixing and stirring behavior of the model system. Namely the ratio of the tracer filament arrest scale (Batchelor scale) to the width of the chaotic region, the rate of dispersal of closely spaced fluid parcels (passive tracers), and the Nakamura effective diffusivity along with the volume integrated tracer variance function. It becomes evident, that chaotic advection introduced by a deterministic disturbance can largely enhance mixing at intermediate times and even for longer times. Many aspects of the underlying mechanisms that lead to these results are discussed and it is found that the deterministic disturbance is especially important for shallow eddies with a large area called "chaotic sea" within the Poincare map.

The article is very well written and touches many important aspects of the intricate problem of the interaction of chaotic advection and diffusive processes. I do recommend the article for publication after some very minor revisions and I really enjoyed reading it. Besides some typography issues and some clarification proposals to figures and definitions I have also some general questions. I hope the authors response might deepen my comprehension and maybe the reply aids to improve the readability of the very nice text even further.

In chronological order (more or less):

Line 69: I would rather talk about important characteristic for the problem considered instead of a "person". Something like: *The terms "important" and "relevant" are somewhat subjective, and a particular aspect, such as the existence of barriers, that is of interest for one scientific question may not be so to another.*

Line 244: though

Fig.2/Fig.3: The scale of the upper row of figures looks not equidistant in my viewer. It is also confusing that the lower plots have a scaling range and the upper ones do not. Further it is said in the caption that the simulations are from another work while in the

text the simulations are claimed to be done within this study. This is a bit confusing.

Line 252: Maybe I missed it but was the chaotic layer thickness introduced before being used here? I feel like maybe it is necessary to revise the usage of chaotic sea, width of chaotic sea, chaotic layer and chaotic region, see also line 16 in the abstract. A mere note would help easier reading.

Line 264: Should it not be equation (13)?

Line 273: Does it make sense to talk of a timescale of a non-dimensional timestep? In general I was sometimes confused by the varying usage of non-dimensional units and dimensional ones. Maybe it could be helpful to use only adimensional ones in the text and make a little table for the conversion?

Line 319: Cauchy-Green.

Line 341: $\gamma = \sigma$?

Line 349: I feel like there is some other reference missing (or not?). Did not Okubo just study up to a scale of 100km in that report?

Fig.5, caption: Which $\delta$ values? (22) or (15)? Even though it becomes clear studying the text it would be easier if it was stated. Just to be curious: What is the result for (15)?

Line 364: Fig. S 2-3

Line 366: 10-20 rotations <-are those the integration times for the FTLE?

Line 377: "rotating can model" is it the standard name for the model used? Why not introduced before? It difficultates understanding using a new name.

Line 421: pf→of

Line 507: Year of citation of Shuckburgh, E. and Haynes, P. should be 2003.

Line 545: The numerical simulations are run using the solver NEK5000 . . .

Fig. 8: $X_0$ instead of $\epsilon$ in caption and in legend the x is small? Further I would find it really helpful for the comparison of the temporal evolution and the peaks of the different cases to have a grid in the background of the plots.

Line 596: After "dye contours" put reference to Fig 10(e)?

Line 599: *A caveat is that $k_{eff}$ is a nonlocal property, the value of which at any point in space and time is influenced by processes occurring at all other locations having the same C at a distinct time t (?).* This time dependence could also be explicitly noted in equation (25) to clarify. I must admit that I am still a bit confused about the details of the calculations of $k_{eff}$ here. As I understand it the $k_{eff}$ values are calculated at each time step from volumes of equal concentrations (and $dC/dV$ is taken from the cumulative distribution of volumes with concentrations $c <= C$ as in the reference [Shuckburgh and Haynes, 2003]). One then gets a function $k_{eff}(C,t)$. The plots of $k_{eff}$ at fixed instances t would thus just be derived by finding the corresponding value of $k_{eff}$ at that $C(t)$? Or is there any such thing as an equivalent latitude used here? I wonder why the two plots of $k_{eff}(t)$ and $C(t)$ look so different. The $k_{eff}$ looks much more noisy. How does $k_{eff}(C, t = 39)$ look like?

Line 607-608: Do the results depend on the details of the initial condition? What must be met to ensure that the results are independent?

Line 631: *Thus the combined effect of smaller diffusivity and finer filaments (i.e., stronger tracer gradients) leads to more rapid mixing across tracer contours.* This is my favorite insight of the analysis and it is, at first glance, counterintuitive to me and still makes so much sense.

Line 667: Here I am confused about the statement: A smaller diffusivity leads to less mixing ($X^2$). Especially when compared to details of Fig. 8.

Line 681: I feel that pros and cons is a bit colloquial and would suggest advantages and disadvantages (or so). But that is just my personal opinion.

As a general remark to the Figures and their descriptions, there is quite some mixed usage of either left-middle-right-bottom-top and a,b,c,d,.... Maybe it would become easier to read just using a,b,c,d,... Especially Fig.'s 8/9/10.

On a similar vein I also find the usages of the word turbulence, turbulent diffusivity, diffusion eddy diffusion and similar expressions confusing too. I would try to avoid using these concepts interchangeably and only use those expressions that were defined before.

---

## Referee Comment (RC3) · Anonymous Referee #3 · 14 Jan 2019

**1  Summary**

The authors study the effect of chaotic advection and diffusion in a submesoscale eddy within a surface mixed layer of the ocean. The eddy is generated using two methods, a three dimensions eddy model and a direct Navier–Stokes simulation. In both cases the flow is generated in a rotating cylinder driven by an off-center lid to break the symmetry. The analysis present three different methods:

- from the Lagrangian Scale ($\delta$), which defines the thinnest filaments that can form from the balance of advection and diffusion, the authors show that advection dominate in the wider chaotic sea region where the layer width is larger than $\delta$.

[Figure]

- from a set of trajectories evolved from the background flow affected two types of perturbation (deterministic and stochastic). Analysis of particle dispersion is presented for different magnitudes and combination of those perturbations. Stochastic perturbation takes over deterministic perturbation over time, an important factor not considered using the scaling arguments of section 1.

- using the Nakamura Effective Diffusivity and the volume integrated tracer variance, the authors give ,quantitive measure on the importance of stirring for different perturbation combinations. Both tools show similar results, when filaments are created, the tracer variance increases until diffusive mixing takes over and wipes every filament. The analysis first focus on integrated values (Fig. 7) then figs. 8-9 present in-depth structures of the flow from dye and effective diffusivity contour.

In general, the paper is well written, the analysis and the description is exhaustive, and easy to follow. The discussion around each of the 3 metrics are well linked together and create a great comparison that will be useful for future research. The topic is important to understand mixing processes in ocean flow in general and could help parametrize model sub-grid phenomena. I would recommend the publication of this paper in NPG with minor revisions. I have divided my comments into two sections, the first one regroups general comments about the analysis and the second are minor corrections and typos.

**2  Comments**

- The methods are really well described but I believed some of the sections are too long. I would like the authors to focus on results and maybe relayed some of the methodology details to Appendix. For example, sec. 3-3.1 and add details to the

results section in 3.2. Section 5 is also a bit hard to follow as the reader is asked to compare two figures, maybe a reorganization of the figures could improve the readability.

- l55-60: the authors should take a look at *Material barriers to diffusive and stochastic transport* by G. Haller, D. Karrasch and F. Kogelbauer, which seeks transport barriers with no diffusion of tracers across it. Those structures could help extract the different regions analyzed in figure 11.

- l575-580: is there a difference between $x_0$ and $X_0$, and later in the caption of Fig.8 there is also an $\epsilon$ ?

- l771-774: 3D dye released and tracking seem almost like an impossible task, especially when the tracers have to be followed for multiple days. Right now, none of those methods can be calculated with observation data. Is there any plan for the applications of such analysis to 3D model outputs (ECCO, HYCOM, etc.) that assimilate data from floats, drifters, CTD casts, etc. ? An analysis performed for example on a Loop Current or Agulhas Current eddy would be interesting.

**3 Typos**

- l140: the streamfunction is not intuitive at all, I believed a sketch of the flow in section two could improve the readability.

- l299: for a for a

- l319: Caushy-Green

- l421: move out pf

- l445: a scale with square root of time could be included in the left panels figure 7

- eq20: I believe it should be $\sigma$ instead of $\gamma$ ?

---

## Referee Comment (RC4) · Anonymous Referee #4 · 18 Jan 2019

This is an ambitious, and in many ways, very good paper. The authors seek to elucidate the competition between chaotic advection and turbulent diffusion in an idealized model of a 3d eddy. Two types of numerical simulation are presented. The first type (I) makes use of an analytically prescribed steady 3d incompressible flow. For this type (I) of simulation two different parameterizations of turbulent diffusion are considered: a) a scale-dependent eddy diffusivity; and b) a stochastic additive noise perturbation to the Lagrangian equations of motion. The second type (II) of simulation makes use of a numerical simulation to the NS equation together with a numerical solution to the advection diffusion equation. Type I and type II simulations could not be more different. The type I simulations are based on some really strong assumptions: 1) there is a separation of scales between the background flow and the turbulent perturbation;

2) the background flow is steady; and 3) the turbulence can be parameterized as a diffusive process (in type I simulations there is no actual turbulent flow). None of these assumptions apply to the type II simulations. Indeed, there is no guarantee that the type II simulations approach anything resembling a quasi-steady flow. With this fundamental difference in mind, I find it remarkable that the qualitative agreement between the type I and type II simulations is as good as it is. Evidently, the authors' analytically prescribed kinematic flow does a really good job of capturing the important underlying physics that are tied to the flow geometry considered. That is impressive. But this good qualitative agreement serves to highlight the importance of assumptions 1) and 2) (maybe also 3)) listed above. It is my view that the validity of almost all of the conclusions/inferences presented in the paper is limited to flows for which assumptions 1) and 2) (again, maybe also 3)) apply. Assumption 2) is both especially important and especially restrictive, and this should be stated clearly by the authors. It is assumption 2) – together with incompressibility – that leads to the underlying structure of a 2 dof autonomous Hamiltonian system. In other words, without assumption 2) there is no basis for the entire chaotic advection paradigm (nested tori, chaotic seas, resonant bands, Poincare sections, etc.) on which the paper is built. Thus, while I found many of the the authors' inferences/conclusions interesting and insightful, I think that the validity of those conclusions/inferences is limited to this class of flows. If the authors disagree, they should explain why. Two additional minor quasi-technical points are worth mentioning. First, in many cases throughout the text the authors make statements about exponential stretching, exponential growth, positive Lyapunov exponents, etc. The authors should be a bit more careful about saying that these behaviors are average behaviors for long times. Second, for the type Ib simulations, the algorithm that the authors use to integrate the stochastic differential equation, Eq 23, is quite crude; there are simple explicit integration schemes for stochastic differential equations (SDEs). (Does the mystery factor of 1/3 described on lines 405-407 somehow account for the fact that a Runge-Kutta algorithm is trying to mimic a real SDE integrator?) Overall, the paper is good and well written. I recommend publication, but I urge

the authors to make some small changes consistent with the remarks above.

---

## Author Comment (AC1) · 12 Mar 2019

**Author response to reviewer comments on "Competition between Chaotic Advection and Diffusion: Stirring and Mixing in a 3D Eddy Model"**

To begin, we would like to thank all reviewers for their close reading and appreciative comments on this paper. It was quite gratifying to receive so many positive remarks. Below, we address the individual comments of each reviewer. Our responses are in italics. Line numbers in our responses refer to the revised paper draft (supplement), while those in the reviewer comments refer to the discussion paper version.

Reviewer 1

1. A sketch of the flow would be helpful when first discussed in detail, showing both

the rotating flow, the Ekman layers, and the off-center perturbation.

*A sketch of the flow was also requested by reviewer 3. A new first figure has been created for this request, which is the first figure attached. It shows the 3D cylinder with the direction of flow and labels indicating Ekman layers. Caption: Sketch of the qualitative velocity field, Eqn.s 7-9. Ekman layers at the top and bottom are where flow has a larger radial component. $\Omega$ is the rotation rate of the system. $X_0$ is the offset between the lid and cylinder rotational centers, as set for the Navier-Stokes simulations.*

2. The Batchelor scale is a little difficult to follow when first presented at the beginning of section 3, but then there is an outstanding discussion of this in section 3.1 (on p. 11) with ample references. Perhaps move that discussion a little earlier.

*Reviewers 2 and 3 also requested the start of this section be cleaned up. In the revised paper we have re-arranged section 3. We have moved the Batchelor scale discussion forward to the beginning of section 3 (lines 253–283), and we have moved the discussion of dimensionalization from the beginning of section 3 to after Okubo's diffusivity is introduced in section 3.1, which is where it becomes relevant.*

3. I will admit that by the time I got halfway through section 5, I was beginning to fade. Although I think that the detail is useful overall, perhaps some trimming of section 5 would be helpful to the readability of the paper. But this isn't critical – it's fine if the authors leave it as is.

*Reviewer 3 also commented that section 5 was challenging. This was indeed a difficult section to write and was the target of a number of revisions and reduction prior to submission. We took another look at it and feel that it would be difficult to condense further without sacrificing important content, but we went through and made a number of minor changes intended to make it a little easier to read:*

*Line 550: $X_0$ is clarified: "The symmetry–breaking perturbation is created by moving the central axis of the imposed surface lid stress a fraction of the radius $X_0$ from the*

cylinder axis"

Line 565: Figure 9 reference now specifies panels a–c

Line 589: Changed "see Figure 10(h))" to "look ahead to Figure 10(h)"

Line 598 and Fig. 11 caption: We now include the diffusivity, $k$, in the description of the case we discuss.

Line 641 Changed "$\kappa_{eff}/k$" to "as measured by $\kappa_{eff}/k$"

We also changed the color scheme of Figs 9-10 (10-11 in the revised version) and improved the quality of these figures to allow more accurate reading of the values.

Reviewer 2

1. Line 69: I would rather talk about important characteristic for the problem considered instead of a "person". Something like: The terms "important" and "relevant" are somewhat subjective, and a particular aspect, such as the existence of barriers, that is of interest for one scientific question may not be so to another.

Changed to: The terms "important" and "relevant" are somewhat subjective, and a particular aspect, such as the existence of barriers, that is of interest for one scientific question may not be so for another.

2. Line 244: though

Changed, thank you! (Now on line 250)

3. Fig.2/Fig.3: The scale of the upper row of figures looks not equidistant in my viewer. It is also confusing that the lower plots have a scaling range and the upper ones do not. Further it is said in the caption that the simulations are from another work while in the text the simulations are claimed to be done within this study. This is a bit confusing.

The non-equidistant ticks in the top panels of Figs. 3 and 4 were due to problems in changing image format. In the revision, we have used higher-quality images and

*removed the ticks.*

*The difference between the top and bottom panels in Figs 3 and 4 are that the top panels were computed using velocities from a dynamically-consistent numerical model, whereas bottom panels were computed using velocities from our analytic (non-dynamically consistent) model. This is now clarified in the caption and in the text. The purpose of these plots was to show that our analytic model (bottom panels) correctly captures the qualitative features of the dynamically-consistent solution (top panels).*

*In this paper we did not re-run the dynamically-consistent NS solver to get the top panels; we simply took these images from P2014. This is now clearly stated in lines 229-230 of the text. Since P2014 only computed Poincare sections but did not compute FTLEs, we have no color images in the top panels. For our analytic model in the bottom panels, however, we have computed both Poincare sections for comparison with the top panels, as well as FTLE fields, whose values we show using the colorbar. These FTLE values will be used in section 3 to compute the Batchelor scales.*

*The new caption for the first of these figures (3) now reads: "Structures in the kinematic model and dynamical simulation for Ekman numbers of 0.25 (a)(c) and 0.125 (b)(d). Top, (a)(b): Poincare maps from P2014 (their Fig. 10), resulting from a dynamically consistent numerical simulation. Bottom, (c)(d): Poincare maps (black) and largest FTLEs (color) resulting from our non-dynamically consistent kinematic analytic model, with $\epsilon = 0.01$ and $x_0$ either -0.5 (c) or -0.9 (d); in color, maximum FTLEs calculated for the kinematic model with integration time 400. In (d), red oval approximately separates the resonant and regular layers (inside) from the chaotic sea region (outside), with the blue line segment showing the width of the chaotic sea. The blue diamond shows the width of an island, which is also the width of the resonant layer."*

4. Line 252: Maybe I missed it but was the chaotic layer thickness introduced before being used here? I feel like maybe it is necessary to revise the usage of chaotic sea, width of chaotic sea, chaotic layer and chaotic region, see also line 16 in the abstract.

A mere note would help easier reading.

*Reviewer 1 also found the start of section 3 to be a bit confusing. Now, near the beginning (lines 260-268), we have: "If $\delta$ is larger than the structures in the flow induced by chaos, then diffusion will overcome advection and wipe out these structures. The structures of interest, induced by the deterministic, symmetry–breaking perturbation (see Fig.s 3–4) are the bands of chaos, called resonant layers, surrounding regular island chains (see blue diamond in Fig. 3d), and the chaotic sea region (outside the red oval in Fig. 3d) located near the cylinder perimeter and central axis, which are identified by visual inspection of Poincare sections. When we compare $\delta$ to these structures, we define their widths as the difference between distances from the central orbit, (r,z)=(0.5,0.5), to the outermost/innermost part of the structure, measured in Poincare sections like Fig.s 3-4."*

5. Line 264: Should it not be equation (13)?

*In fact, it should be equation (9), for the background azimuthal velocity of the kinematic model. This has been changed.*

6. Line 273: Does it make sense to talk of a timescale of a non-dimensional timestep? In general I was sometimes confused by the varying usage of non-dimensional units and dimensional ones. Maybe it could be helpful to use only adimensional ones in the text and make a little table for the conversion?

*Thank you for your suggestion on making the text more readable. We have added Table 1, figure 2 below, with the correspondence between dimensional and nondimensional values, and removed references to dimensional values of model variables in sections 3-5 (e.g. lines 364,367,461,463).*

7. Line 319: Cauchy-Green.

*Changed, thank you.*

8. Line 341: $\gamma = \sigma$ ?

*Several errors existed in this equation, which is derived in Appendix B. It is now corrected.*

9. Line 349: I feel like there is some other reference missing (or not?). Did not Okubo just study up to a scale of 100km in that report?

*You are correct, we have updated the upper limit (line 332).*

10. Fig.5, caption: Which $\delta$ values? (22) or (15)? Even though it becomes clear studying the text it would be easier if it was stated. Just to be curious: What is the result for (15)?

*These values are computed with Eqn. 22, which is now indicated in the caption. With Eqn. 15, the Batchelor scale is similar to or larger than the layer widths in all cases.*

11. Line 364: Fig. S 2-3

*These are in fact referencing figures (plural) 2 and 3, not supplementary figures. This matter has been corrected in the revision (line 363).*

12. Line 366: 10-20 rotations <-are those the integration times for the FTLE?

*The reviewer is correct. We have clarified the sentence in question as follows: "FTLEs were estimated over an integration time of 400; the range of FTLE magnitudes does not noticeably change when the integration time is decreased by half."*

13. Line 377: "rotating can model" is it the standard name for the model used? Why not introduced before? It difficultates understanding using a new name.

*This now reads "kinematic model"; thank you for pointing this out.*

14. Line 421: pf→of

*This has been changed, thank you.*

15. Line 507: Year of citation of Shuckburgh, E. and Haynes, P. should be 2003.

*In this case, we are citing Haynes and Shuckburgh's work on the application of the Nakamura effective diffusivity to the atmosphere, which is described in a pair of papers in JGR in 2000, not the 2003 Shuckburgh and Haynes work on the quantitative applicability of the effective diffusivity in Physics of Fluids in 2003. (The latter is also relevant for our paper and is cited on line 519.)*

16. Line 545: The numerical simulations are run using the solver NEK5000...

*We have added "the solver", thank you.*

17. Fig. 8: X0 instead of epsilon in caption and in legend the x is small? Further I would find it really helpful for the comparison of the temporal evolution and the peaks of the different cases to have a grid in the background of the plots.

$X_0$ *has replaced $\epsilon$ and $x_0$, thank you. We have also added a grid.*

18. Line 596: After "dye contours" put reference to Fig 10(e)?

*We have referenced Fig 10i, thank you for demonstrating how confusing this can be without that reference.*

19. Line 599: A caveat is that k eff is a nonlocal property, the value of which at any point in space and time is influenced by processes occurring at all other locations having the same C at a distinct time t (?). This time dependence could also be explicitly noted in equation (25) to clarify. I must admit that I am still a bit confused about the details of the calculations of k eff here. As I understand it the keff values are calculated at each time step from volumes of equal concentrations (and dC/dV is taken from the cumulative distribution of volumes with concentrations c < = C as in the reference [Shuckburgh and Haynes, 2003]). One then gets a function k eff ( C,t ). The plots of k eff at fixed instances t would thus just be derived by finding the corresponding value of k eff at that C (t)? Or is there any such thing as an equivalent latitude used here? I wonder why the two plots of keff(t) and C(t) look so different. The keff looks much more noisy. How does keff(C,t= 39) look like?

*You are correct about the calculation of keff. The plots of keff are noisier than those of C because the function keff(C) is not very smooth: small variations in C can correspond to larger, non-monotonic changes is keff. This is now addressed in the text lines 600-604: "A caveat is that keff is a nonlocal property, so plots show the values keff(C) mapped onto the locations on the sections with corresponding dye concentrations, C, while they are calculated using the distribution of C over the whole volume at that time. These mappings are noisier than the sections of C because the numerically computed keff(C) is nonmonotonic and can have large changes with small changes in C."*

20. Line 607-608: Do the results depend on the details of the initial condition? What must be met to ensure that the results are independent?

*In our simulations, we see a transition from the large-scale vertical gradient in tracer concentration to approximately toroidal tracer distribution after several overturning cycles. For an initial broad gradient in any direction, we expect the same realignment after the first few overturnings as the tracer is passively advected by the background velocity field. We believe, then, that the tracer distribution that exists at later times is somewhat independent of initial distribution. We have added this idea to the text, lines 615–620, as: "With no disturbance present $(X_0 = 0)$ and $k = 10^{-4}$ (Fig. 10(b)), the initially horizontal lines of constant C have been advected by the axially symmetric overturning circulation such that contours of constant C are roughly aligned with the overturning streamfunction. For an initial broad gradient in any direction, we expect the same realignment after the first few overturnings as the tracer is passively advected by the background velocity field. We believe, then, that the tracer distribution that exists at later times is somewhat independent of initial distribution." It would be interesting to test this, but one would have to do it over a range of tracer distributions. Our paper is quite long already, so this may have to be something to explore in the future.*

21. Line 631: Thus the combined effect of smaller diffusivity and finer filaments (i.e., stronger tracer gradients) leads to more rapid mixing across tracer contours. This is my favorite insight of the analysis and it is, at first glance, counterintuitive to me and

still makes so much sense.

*Thank you.*

22. Line 667: Here I am confused about the statement: A smaller diffusivity leads to less mixing (X2). Especially when compared to details of Fig. 8.

*We agree with the reviewer that the wording of this sentence was indeed ambiguous. We have rewritten the sentence in question as follows (lines 680–681): "A smaller diffusivity leads to more filamentation (higher $^2$) and highly elevated enhancement (much larger $\kappa_{eff}$)."*

23. Line 681: I feel that pros and cons is a bit colloquial and would suggest advantages and disadvantages (or so). But that is just my personal opinion.

*We have changed this line, thank you for marking it.*

24. General comment 1: As a general remark to the Figures and their descriptions, there is quite some mixed usage of either left-middle-right-bottom-top and a,b,c,d,.... Maybe it would become easier to read just using a,b,c,d,... Especially Fig.'s 8/9/10.

*In the text we have replaced references by location to references by panel letters. In the actual captions of figures, we use positions of rows and columns in order to indicate the pattern of the panels- rows and columns generally have the same type of data or data from the same simulation shown. However, we now also note the panel letters to avoid confusion.*

25. General comment 2: On a similar vein I also find the usages of the word turbulence, turbulent diffusivity, diffusion, eddy diffusion and similar expressions confusing too. I would try to avoid using these concepts interchangeably and only use those expressions that were defined before.

*We have tried to clean up our lexicon. Both terms "eddy diffusion" and "eddy diffusivity" have been eliminated. We now consistently use terms "chaotic advection" and "turbu-*

*lent diffusion" to refer to the two dominant processes that influence tracer evolution. For example, the new abstract now reads:*

*"The importance of chaotic advection relative to turbulent diffusion is investigated in an idealized model of a 3D swirling and overturning ocean eddy...Turbulent diffusion is alternatively represented by: 1) an explicit, observation–based, scale–dependent diffusivity, 2) stochastic noise, added to a deterministic velocity field, or 3) explicit and implicit diffusion in a spectral numerical model of Navier–Stokes equations."*

Reviewer 3

General comment 1: The methods are really well described but I believed some of the sections are too long. I would like the authors to focus on results and maybe relayed some of the methodology details to Appendix. For example, sec. 3-3.1 and add details to the results section in 3.2. Section 5 is also a bit hard to follow as the reader is asked to compare two figures, maybe a reorganization of the figures could improve the readability.

*Reviewer 1 agreed that the structure of section 3 was not optimal and that section 5 was challenging. We have re-structured and streamlined section 3. Specifically, we have moved the Batchelor scale discussion forward to the beginning of section 3, and the discussion of dimensionalization from the beginning of section 3 to after Okubo's diffusivity is introduced in section 3.1.*

*Section 5 was indeed difficult to write and was the target of a number of revisions and reduction prior to submission. We took another look at it and feel that it would be difficult to condense further without sacrificing important content, but we went through and made a number of minor changes intended to make it a little easier to read:*

*Line 550: $X_0$ is clarified: "The symmetry–breaking perturbation is created by moving the central axis of the imposed surface lid stress a fraction of the radius $X_0$ from the cylinder axis."*

*Line 565: Figure 9 reference now specifies panels (a)—(c)*

*Line 589: Changed "see Figure 10(h))" to "look ahead to Figure 10(h)"*

*Line 598 and Fig. 11 caption: We now include the diffusivity, k, in the description of the case we discuss.*

*Line 641 Changed "$\kappa_{eff}/k$" to "as measured by $\kappa_{eff}/k$"*

*We also changed the color scheme of Figs 9-10 (10-11 in the revised version) and improved the quality of these figures to allow more accurate reading of the values. Finally, we now consistently refer to the subpanels in the multi-panel figures by their letters (a,b,...) instead of their location (middle, left, etc.).*

2: Lines 55-60: the authors should take a look at "Material barriers to diffusive and stochastic transport" by G. Haller, D. Karrasch and F. Kogelbauer, which seeks transport barriers with no diffusion of tracers across it. Those structures could help extract the different regions analyzed in figure 11.

*We thank the reviewer for the reference to a relevant paper and for the suggestion regarding figure 11. While we agree that it would be interesting to try re-doing the calculations presented for the slightly different regular region identified via this other technique, the resolution of the tracer field in our existing simulations is too low to allow for a precise calculation of Keff within that exact island. Note that we only use Poincare sections as guidance for picking red and blue regions with qualitatively different behavior. We have added the following sentence to address this issue near the end of section 5, line 667: "While we used Poincare sections as guidance for defining regular and chaotic regions, other methods (for example, Haller et al., 2018) could be used instead for the more precise delineation of the phase space."*

3: Lines 575-580: is there a difference between x0 and X0, and later in the caption of Fig.8 there is also an epsilon?

*Yes, there is a difference. As noted in the first paragraph of section 2.2 (l215-220), x0*

*is a parameter for the offset in the perturbation for the kinematic model, while X0 is the offset in the NS simulation. Epsilon is the strength of the perturbation for the kinematic model. In the caption of figure 8 the epsilon should be X0; this has been corrected. We have also labeled X0 in the new figure (now Figure 1) requested by the first reviewer.*

4: Lines 771-774: 3D dye released and tracking seem almost like an impossible task, especially when the tracers have to be followed for multiple days. Right now, none of those methods can be calculated with observation data. Is there any plan for the applications of such analysis to 3D model outputs (ECCO, HYCOM, etc.) that assimilate data from floats, drifters, CTD casts, etc.? An analysis performed for example on a Loop Current or Agulhas Current eddy would be interesting.

*We thank the reviewer for the comment and suggestion. We agree that any application to the ocean in 3D is extraordinarily challenging for any of our measures. It would be quite interesting to apply our analysis to data-assimilative models, but that is outside the scope of our work. There is also the question of whether any of the cited models can reproduce the vertical velocity field correctly. Often, a certain degree of horizontal averaging is needed to get a vertical velocity that looks sensible.*

5. Line 140: the streamfunction is not intuitive at all, I believed a sketch of the flow in section two could improve the readability.

*To better relate the streamfunction to the velocities, we have added at its introduction (line 145) "The streamfunction relates to the velocities by the negative z-derivative of $\Psi$ being the radial velocity and the radial derivative being the vertical velocity." A sketch of the flow was also requested by reviewer 1 and is now presented as the first figure- it is included in the response there, the first one in this document.*

6. Line 299: for a for a

*Now one for a, thank you.*

7. Line 319: Caushy-Green

*Cauchy, thank you.*

8. Line 421: move out pf

*Of, thank you.*

9. Line 445: a scale with square root of time could be included in the left panels of figure 7

*The square root of time curve has been added to panel (a).*

10. Equation 20: I believe it should be $\sigma$ instead of $\gamma$ ?

*Several errors existed in this equation, which is derived in Appendix B. It is now corrected.*

Reviewer 4

Here we paraphrase the comments in order to shorten them and address each individually.

1. The kinematic model assumes 1) there is a separation of scales between the background flow and the turbulent perturbation; 2) the background flow is steady; and 3) the turbulence can be parameterized as a diffusive process (in the simulations of this model there is no actual turbulent flow). It is my view that the validity of almost all of the conclusions/inferences presented in the paper is limited to flows for which assumptions 1) and 2) (again, maybe also 3)) apply. None of these assumptions apply to the NS simulations. Indeed, there is no guarantee that those simulations approach anything resembling a quasi-steady flow. Assumption 2) is both especially important and especially restrictive, and this should be stated clearly by the authors. It is assumption 2) – together with incompressibility – that leads to the underlying structure of a 2 dof autonomous Hamiltonian system.

*You are correct that a separation in spatial and temporal scales between the background flow and the turbulent perturbation is necessary for the analysis we perform.*

*This is explicitly stated in the introduction (lines 58–68): "The relevance of chaotic advection for the stirring of material within geophysical flows would appear to rest on several criteria. The first is that the flow field contain persistent, long–lived (on the time–scale of interest) features such as gyres, eddies and jets, that by themselves generate regions of elevated stirring as well as separating barriers. Secondly, the stirring within these regions should be at least as important as that due to smaller scale, intermittent features (i.e. small scale turbulence). Third, the barriers that exist in the absence of small–scale turbulence should retain meaning as suppressors of exchange between the rapidly–stirred regions in the presence of the small–scale turbulence. For the flow considered in this paper the first aspect has been investigated and shown to be true (citations); this work concentrates on investigating the second and third aspects."*

*As for the NS simulations, they do reach a nearly steady flow, as have laboratory experiments with similar setups (e.g. Fountain et al. 2001). In our work, NS simulations produced a circulation that was observed to be steady over hundreds of azimuthal cycling times, as evidenced by the replication of periodic orbits in the stroboscopic sections in the top panels (labelled a,c) of our figs. 3 and 4 (previously 2 and 3). Please also see Pratt et al., 2014 for more examples of NS-based Poincare sections over a wide range of parameters.*

*We have added to the conclusion (lines 792-794) "Although the focus of this current paper is on the behavior of a steady 3D eddy flow subject to a turbulent diffusion, similar results are expected to hold for 3D eddy flows with time-periodic and time-quasiperiodic behavior." To expand on that idea, note that treatment of a time-periodic 3D eddy flow (but without the focus on advection vs diffusion) is described in Rypina et al. (2015), who used theoretical arguments, analytic kinematic model and NS simulations to study Lagrangian transport and transport barriers arising in a time-dependent 3d idealized eddy. We could have carried out the same suite of experiments for that time-dependent flow as here, invoking the same assumptions of time scale separation and parameterization of turbulence as a diffusive process. So one does not lose the*

*features induced by chaotic advection just because of time dependence. Our hope is to analyze time-dependent and otherwise more complex flows in the future using the presented set of tools, and to inspire others to consider our measures of the relative impacts of chaotic advection and turbulence or diffusion in other flows. Note also that a flow that varies on a significantly longer timescale than more intermittent perturbations could be amenable to analysis similar to the steady case.*

2. In many cases throughout the text the authors make statements about exponential stretching, exponential growth, positive Lyapunov exponents, etc. The authors should be a bit more careful about saying that these behaviors are average behaviors for long times.

*We agree with the reviewer and in the revised paper we now avoid such non-rigorous statements. Instead, we use terms akin to "fast chaotic-advection-induced separation/stretching" (lines 445, 482, for example) and phrases like "average exponential growth, as measured by FTLEs" (lines 256, 286, 440, 745). We have also rewritten the sentence introducing FTLEs as follows (lines 292-293): "The FTLE quantifies the average exponential separation rate between a trajectory and its close neighbors over a finite time interval."*

3. For the trajectory integrations in the kinematic model with added stochasticity, the algorithm that the authors use to integrate the stochastic differential equation, Eq 23, is quite crude; there are simple explicit integration schemes for stochastic differential equations (SDEs). (Does the mystery factor of 1/3 described on lines 405-407 somehow account for the fact that a Runge-Kutta algorithm is trying to mimic a real SDE integrator?)

*Yes, we used a very simple method for integrating a stochastic process. Other integration schemes might provide a more accurate solution. However, we carefully tested that its mean and standard deviation behaved properly over time without a background velocity before applying it. The stochastic perturbations are added to the deterministic*

*velocity at each full timestep. The 1/3 factor is due to the fourth-order integration function, which estimates the next point using the weighted sum of estimates of the velocity at the current position (v1, weight 1/6), the halfway point estimated from the current position (v2, weight 1/3), the halfway point estimated using v2 (v3, weight 1/3), and the final point estimated using v3 (v4, weight 1/6). Only v1 and v4 include stochastic additions, leading to the 1/3 factor. We have added the following at lines 393–408 to clarify: "Using the described stochastic perturbation, although it is quite simple, with $U_{bi} = 0$ or a constant, the variance of a set of trajectories grows linearly in time, while the standard deviation grows linearly with the square root of time, as expected for diffusion... This diffusivity requires a certain step size s for the stochastic perturbation, which relates to the distribution of $u'$ by $s = \sigma \Delta t/3$, with $\sigma$ the standard deviation of $u'$, $\Delta t$ the numerical timestep (0.01), and the factor of 3 due to the details of a fourth–order Runga–Kutta integration. The next position, using this method, is estimated using the weighted sum of estimates of the velocity at the current position (v1, weight 1/6), the halfway point estimated from the current position (v2, weight 1/3), the halfway point estimated using v2 (v3, weight 1/3), and the final point estimated using v3 (v4, weight 1/6). Only v1 and v4 include stochastic additions, leading to the 1/3 factor."*

Please also note the supplement to this comment:
https://www.nonlin-processes-geophys-discuss.net/npg-2018-54/npg-2018-54-AC1-supplement.pdf

———————————————

W

cylinder rotation axis

lid rotation axis

$X_o$

Ekman layer

Ekman layer

**Fig. 1.** Sketch of the qualitative velocity field, Eqn.s 7-9. Ekman layers at the top and bottom are where flow has a larger radial component. $\Omega$ is the rotation rate of the system. X0 is the offset between the

| Variable | Nondimensional Value | Scaling factor | Dimensional value, E=0.25 | E=0.125 | E=0.02 | E=0.0005 |
|---|---|---|---|---|---|---|
| $\delta_E$ | $a/\sqrt{E}$ | 40m | 40m | 40m | 40m | |
| $H$ | 1 | $a\delta_E/\sqrt{E}$ | 80m | 113m | 283m | 1789m |
| $u$ | 0.01–0.1 | 1m/s | 0.01–0.1m/s | 0.01–0.1m/s | 0.01–0.1m/s | 0.01–0.1m/s |
| $w$ | $2\cdot10^{-4}$–0.05 | 1m/s | 0.005–0.05m/s | 0.003–0.04m/s | 0.001–0.014m/s | $2\cdot10^{-4}$–$2\cdot10^{-3}$m/s |
| timestep | 1 (1000) | $H/1\text{ms}^{-1}$ | 84s (23h) | 113s (31h) | 283s (3.3d) | 1789s (20.7d) |
| winding time | 20–200 | time | 28min–4.6hr | 39min–6.6hr | 1.5–16.5hr | 3.3–33hr |
| overturning time | 30–400 | time | 42min–9h | 1–12h | 2–31h | 15h–8d |
| $\kappa$ | e.g. $10^{-5}$ | $H\text{m}^2\text{s}^{-1}$ | $8\cdot10^{-4}\text{m}^2\text{s}^{-1}$ | $1.1\cdot10^{-3}\text{m}^2\text{s}^{-1}$ | $2.8\cdot10^{-3}\text{m}^2\text{s}^{-1}$ | $1.8\cdot10^{-2}\
[revised manuscript text omitted]